# A Fully Integrated Biorefinery Process for the Valorization of *Ulva fasciata* into Different Green and Sustainable Value-Added Products

Nour Sh. El-Gendy [1,2,*] , Hussein N. Nassar [1,2,3] , Abdallah R. Ismail [1] , Hager R. Ali [1] , Basma Ahmed Ali [4] , Khaled M. Abdelsalam [5] and Manal Mubarak [6]

1 Egyptian Petroleum Research Institute (EPRI), Nasr City, Cairo P.O. Box 11727, Egypt
2 Center of Excellence, October University for Modern Sciences and Arts (MSA), 6th of October City, Giza P.O. Box 12566, Egypt
3 Faculty of Physical Therapy, October University for Modern Sciences and Arts (MSA), 6th of October City, Giza P.O. Box 12566, Egypt
4 General Organization for Export and Import Control (GOEIC), Cairo P.O. Box 11522, Egypt
5 Marine Environment Division, National Institute of Oceanography and Fisheries NIOF, Alexandria P.O. Box 21519, Egypt
6 Soil and Water Department, Faculty of Agriculture, Ain Shamas Unversity, Cairo P.O. Box 11241, Egypt
* Correspondence: nourepri@yahoo.com or nshelgendy@msa.edu.eg; Tel.: +20-1004-510-006

**Abstract:** In the framework of a sustainable marine bioeconomy, the present work describes an advanced, eco-friendly, fully integrated biorefinery process for marine *Ulva fasciata* macroalgae. That would serve as a solution for ecosystem bioremediation, an effective utilization of marine macroalgal resources, and a new initiative to promote a green and low-carbon economy. *Ulva fasciata* biomass can be utilized as an organic fertilizer with total N, $P_2O_5$, and $K_2O$ contents of 3.17% and a C/N ratio of 11.71. It can also be used as a solid biofuel with a sufficient calorific value of 15.19 MJ/kg. It has high carbohydrate content and low lignin content of approximately 44.85% and 1.5%, respectively, which recommend its applicability in bioethanol and biobutanol production. Its protein, fiber, lipid, and ash contents of approximately 13.13%, 9.9%, 3.27%, and 21%, respectively with relatively high concentrations of omega-3 fatty acids (n-3 PUFAs) and omega-9 fatty acids (n-9 MUFAs) and relatively low omega-6 fatty acids (n-6 PUFAs) and a n-6/n-3 ratio of 0.13 also recommend its applicability as food additives and animal feeders. Moreover, the suggested sequential zero-waste biomass residue process yielded 34.89% mineral-rich water extract (MRWE), 2.61% chlorophyll$_{a,b}$, 0.41% carotenoids, 12.55% starch, 3.27% lipids, 22.24% ulvan, 13.37% proteins, and 10.66% cellulose of *Ulva fasciata* dry weight. The efficient biocidal activity of extracted ulvan against pathogenic microorganisms and sulfate-reducing bacteria recommends its application for medical purposes, water densification, and mitigation of microbially induced corrosion in the oil and gas industry.

**Keywords:** *Ulva fasciata*; valorization; biorefinery; blue economy; value-added products; biopolymers

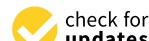



## 1. Introduction

Green macroalgae is abundant on the Egyptian Mediterranean shoreline, and it is left unutilized, although it is used for food purposes in many Fareast Asian countries [1]. They are usually found throughout the year and peak during the summer and beginning of the autumn seasons in shallow water, within low intertidal zones and marshes, on bared rocks, and stagnant tide meres [2,3]. It usually causes eutrophication along enclosed coasts [4–6]. The blooming of macroalgae has strong negative impacts on aquaculture, the marine ecosystem, tourism, and the economy [7–9]. *Ulva fasciata* is from the Chlorophyta phylum, the Chlorophyceae class, the Ulvales order, and the Ulvaceae family. It is also known as sea lettuce and limu palahalaha [10]. It is commonly found attached to intertidal rocks and coastal hard bedrocks, in tide pools, and on reef flats, within low wave forces,

reduced herbivory, and elevated nutrients provide feeding [11]. *Ulva fasciata* is known to be tolerant of stressful conditions, and usually its presence indicates freshwater input or pollution. Besides, it has high capabilities to bioaccumulate minerals and heavy metals, which limit its applicability for human consumption [6].

Valorization of such wasted green macroalgae into different value-added products and sustainable biopolymers has recently attracted a lot of interest [2,12–17] for their wide applications in the dyes, food, pharmaceutical, animal feeders, fertilizers, bioplastics, and biofuels industries. In previously reported fully integrated processes, Indian *Ulva fasciata* biomass was valorized into mineral-rich liquid extract, lipids, ulvan, cellulose, and bioethanol [8]; *Ulva lactuca* was valorized into mineral-rich sap, lipids, ulvan, proteins, and cellulose [18]; *Ulva ohnoi* was valorized into value-added bioproducts; salts, starch, lipids, ulvan, proteins, and cellulose [19]. To our knowledge, there is no reported sequentially fully integrated zero-waste ecofriendly process for the valorization of *Ulva fasciata* into pigments, antioxidants, mineral-rich water extract (MRWE), lipids, and proteins, besides the valuable biopolymers; starch, ulvan, and cellulose.

In an attempt to flourish the marine bio-economy, this work aims to valorize *Ulva fasciata* bloomed biomass polluting the Egyptian Mediterranean shoreline into various value-added sustainable biofertilizer, biofuel, biocide, and bioproducts with various industrial applications, including pigments, antioxidants, mineral-rich water extract (MRWE), starch, lipids, ulvan, proteins, and cellulose, using a cutting-edge environmentally friendly, waste-free, sequential full-integrated biorefinery process. That would help in the achievement of the seventeen goals of sustainable development via opening a new market for sustainable marine biorefineries, and low-carbon marine algal biobased industries, enriching the marine algae blue economy, reinforcing the forthcoming maritime frugalities, and overcoming many of the bottlenecks in the green economy, for example, the food versus fuel issue, deforestation, saving arable lands for food crops, fertilizer and pesticide use, and freshwater consumption.

## 2. Materials and Methods

### 2.1. Sampling of Algal Biomass

The green macroalgal *Ulva fasciata* biomass has been collected from Miami beach, Alexandria Egyptian Mediterranean shoreline during August and October 2022 (31°16′13.36″ N 29°59′11.96″ E) (Figure 1).

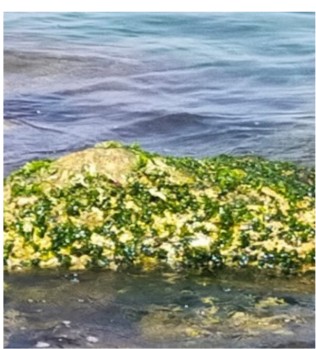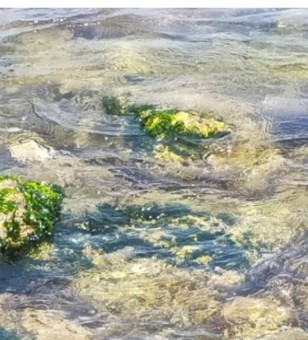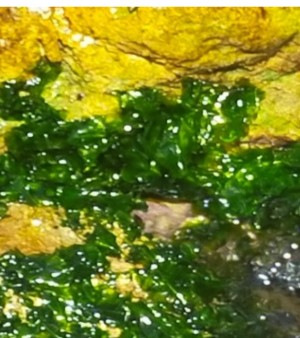

**Figure 1.** *Ulva fasciata* on Egyptian Mediterranean shoreline (31°16′13.36″ N 29°59′11.96″ E).

The morphological identification of the collected macroalgae has been performed according to Ismail and Mohamed [20] by microscopic examination using a stereo microscope (model P-20, Novex, part of EUROMEX microscopes BV, Arnheim, the Netherlands) and a light microscope (model B-192, Optika, S.r.l. Via Rigla, Ponteranica (BG), Italy), at the National Institute of Oceanography and Fisheries (NIOF), Alexandria Branch. Field emission scanning electron microscope embedded smart energy dispersive X-ray spectroscopy (FESEM-EDX, model Sigma 300VP Carl Zeiss, Jena, Germany) has also been used,

according to El-Naggar et al. [21] and El-Sheekh et al. [22], for the macroalgae surface morphological identification at the Egyptian Petroleum Research Institute.

The collected macroalgae were washed with tap water to remove salt, unwanted sand, and any other debris. Then, they were left to dry in the sunlight. Then, they were ground and sieved into homogenized powder (0.8–1 mm). Finally, they were preserved in sealed, clean plastic bags until being used.

### 2.2. Physicochemical Characterization of the Collected Algal Biomass

The nutritional composition of the algal biomass has been determined using the methods described in AOAC [23] at the Soil and Water Unit, Central Lab, Faculty of Agriculture, Ain Shams University. The proximate analysis of the algal biomass has been performed at the General Organization for Export and Import Control (GOEIC), which included measuring the moisture content [24], volatile matter [25], ash content [26], and calorific value [27]. The cellulose, hemicellulose, and lignin contents have also been determined at GOEIC according to the method reported by Moubasher et al. [28]. The biochemical analysis of the collected macroalgae was performed at the Soil and Water Unit, Central Lab, Faculty of Agriculture, Ain Shams University. The organic carbon and organic matter contents were determined according to the Walkley and Black method [29]. The pH and electrical conductivity were measured according to the method reported by Jackson [30]. Total N was determined by the Kjeldahl digestion method, as described by Chapman and Pratt [31], and the conversion factor of 6.25 was used to calculate the crude protein content. Total phosphorus was determined according to the method described by Watanabe and Olsen [32]. Minerals and heavy metals were measured according to the method described by Benton [33] using inductively coupled plasma mass spectrometry (ICP-MS, Spectro Ciros CCD ICP-OES, Spectro Analytical Instruments, Kleve, Germany).

### 2.3. Extraction of Different Value-Added Products from the Collected Macroalgae

The sequential, fully integrated process for the extraction of different value-added products from the collected *Ulva fasciata* is presented in Figure 2. The pigment extraction from *Ulva fasciata* was done according to Wahlström et al. [34]. The extraction of the mineral-rich fraction and starch has been performed according to the methods reported by Trivedi et al. [8] and Prabhu et al. [19], respectively, followed by the extraction of lipid, ulvan, protein, and cellulose in a sequential manner according to the methods reported by Trivedi et al. [8] and Mhatre et al. [35].

### 2.4. Analytical Tools

Pigments were identified according to Ismail [36] by an ultraviolet/visible/near-infrared (UV/Vis/NIR) double beam spectrophotometer (model V-570, JASCO International Co., Ltd., Tokyo, Japan) via the scanning of the characteristic absorption spectra at wavelengths of 190–800 nm [37]. The lipid profile has been identified using a TRACE GC Ultra Gas Chromatographs (THERMO Scientific Corp., Waltham, MA, USA), equipped with a HP-5MS (30 m $\times$ 0.25 mm $\times$ 0.25 $\mu$m) column, and coupled with a thermo mass spectrometer detector (ISQ Single Quadrupole Mass Spectrometer, THERMO Scientific Corp., Waltham, MA, USA) (GC-MS). Briefly, a 2 $\mu$L sample was injected into the GC-MS system at the initial column temperature of 45 °C, held for 1 min, ramped to 150 °C with 4 °C/min, held for 3 min, ramped up to 280 °C with 9 °C/min, and held for another 5 min. The injector was kept at 250 °C in split mode (10:1) with the carrier gas helium. The ion source was maintained at 230 °C. Mass spectrometric measurements were performed using electron impact ionization (EI) at an ionizing voltage of 70 eV, and a scanning range of $m/z$ 50–550. Peak identification was accomplished by comparing mass spectra to the mass spectral library of the National Institute of Standards and Technology (NIST) 2011 (https://www.nist.gov). Using the Bradford method and bovine serum albumin (BSA) as the reference standard, protein concentration was measured [38].

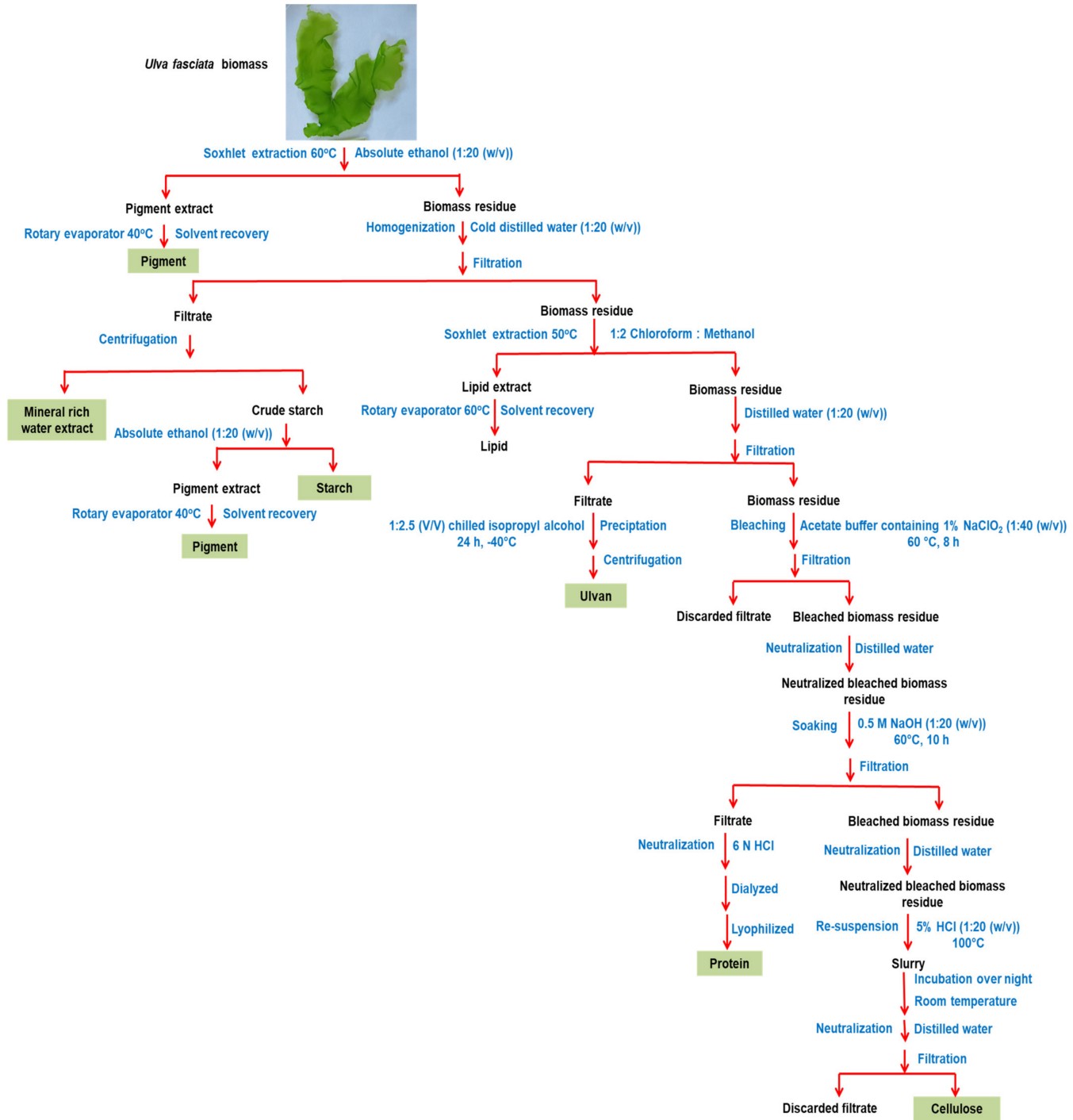

**Figure 2.** A fully integrated process for the extraction of value-added products from the collected *Ulva fasciata*.

The extracted value-added products have been characterized by Fourier transform infrared (FTIR; Perkin Elmer Spectrum One, Waltham, MA, USA), applying the KBr disc method, and the spectra were scanned in the range of 400–4000 cm$^{-1}$ with 4 cm$^{-1}$ and 16 cm/min scan resolution and rate, respectively. The X-ray diffractometer (XRD) patterns of the extracted starch, ulvan, and cellulose have been confirmed using high-resolution XRD (PANalytical XPERT PRO MPD, Lelyweg, EA Almelo, the Netherlands) coupled with a Cu k radiation source (λ = 1.5418 Å) operated at 40 kV and 40 mA. The average particle size was determined by a dynamic light scattering instrument (DLS, ZetaSizer Nano Series (HT), Nano ZS, Malvern Instruments, Malvern, UK). Field emission scanning electron

microscope embedded smart energy dispersive X-ray spectroscopy (FESEM-EDX, model Sigma 300VP Carl Zeiss, Jena, Germany) has been used to study the granule morphology of the value-added products. The thermal properties of the produced value-added products were studied using Q600 SD simultaneous differential scanning calorimetry—thermal gravimetric analysis (DSC–TGA), under nitrogen conditions. $N_2$ was purged at the rate of 100 mL/min, and the temperature was programmed from ambient temperature to 950 °C at the ramp rate of 10 °C/min. The DSC data of the extracted value-added products were obtained using TA instrument Discovery DSC250, New Castle, DE, USA, under nitrogen conditions. $N_2$ was purged at a rate of 100 mL/min, and the temperature was programmed from 0 °C to 250 °C, with a heating rate of 5 °C/min. The obtained thermograms were analyzed using the TA Instruments Trios software (V5.2.2.47561, New Castle, DE, USA), and the gelatinization enthalpy ($\Delta H$ J/g) was calculated by dividing the integrated peak area by the heating temperature rate (5 °C/min) and further by the weight of the value-added product slurry (5 mg). All tests were performed in triplicate.

### 2.5. The Antimicrobial Activity of the Extracted Ulvan

The agar-well diffusion technique [39] was applied to test the antimicrobial activity of *Ulva fasciata* extracted ulvan (5 g/L) against the Gram-positive bacteria (*Bacillus subtilis* ATCC 6633 and, *Staphylococcus aureus* ATCC 35556), Gram-negative bacteria (*Escherichia coli* ATCC 23282 and *Pseudomonas putida* ATCC 10145), yeast (*Candida albicans* IMRU 3669), and filamentous fungus (*Aspergillus niger* ATCC 16404). Sterile distilled water was used as a negative control, while standard antibiotics were used as positive controls: streptomycin (50 μg/mL) as antibacterial and metronidazole (100 μg/mL) as an anti-yeast and anti-fungal. With the use of a sterile gel puncture, 6 mm-diameter agar wells were created, and the zone of inhibition (ZOI mm), which appeared as a clear region around the wells, was recorded. All examinations were done in triplicate and the listed data are the average of the obtained results. The activity index of the extracted *Ulva fasciata* ulvan was calculated on the basis of the size of the inhibition zone (mm) by the following formula:

$$Activity\ index = \frac{Ulvan\ inhibition\ zone\ (\text{mm})}{Standard\ antibiotic\ inhibition\ zone\ (\text{mm})} \tag{1}$$

The minimum inhibitory concentration (MIC) of an antimicrobial agent is the lowest amount that stops microorganisms from growing in a way that can be seen. In liquid, culture medium, extracted ulvan was evaluated at a range of concentrations (50–3000 mg/L). The optical density at 600 nm is measured to determine the amount of microbial growth after the inoculation of the tested strains into each ulvan concentration after 24 h for bacteria and yeast and 72 h for fungi. Loopfuls of the evaluated microbial cultures were placed onto sterile nutrient agar plates using the streak plate technique from all tubes that showed no evidence of growth or turbidity (MIC and higher dilutions). The plates were then cultured for fungi for 72 h at 30 °C and for bacteria and yeast at 30 °C for 24 h. The lowest concentration at which the tested microorganisms did not grow was identified as the minimum lethal concentration (MLC) against the tested microbial strains. In the experiment, there were two types of controls: a positive control (test tubes with inoculum and nutrient broth medium without ulvan) and a negative control (test tubes with ulvan and nutrient broth medium without the inoculum). Each experiment was run in triplicate, and the mean value was computed.

The biocidal activity of the extracted ulvan (500–3000 mg/L) was also tested against a mixed culture of halotolerant planktonic sulfate-reducing bacteria (SRB) of salinity 30,000 mg/L collected from an Egyptian oil field and the standard non-halotolerant strain *Desulfovibrio sapovorans* ATCC 33892. According to Omran et al. [40], postgate medium B was utilized to enrich the planktonic SRB, and the most probable number technique was applied according to ASTM-D4412-84 [41] for the enumeration of the planktonic SRB. Vials were incubated at 30 °C for 21 days, and the growth of SRB was indicated by the

formation of a black ferrous sulfide (FeS) precipitate. The positive control was inoculated vials without ulvan, whereas the negative control was non-inoculated vials with ulvan.

## 3. Results and Discussion

### 3.1. Morphological Examination of the Collected Ulva fasciata

The visual observation (Figure 3a) and the stereo microscope examination (Figure 3b) revealed a pale to dark green elongated, skinny thallus of about 10 cm that is divided into a number of lobes, which are broader at the bottom and narrower at the tip with irregular edges. That matched well with the observations reported by AbouGabal et al. [42].

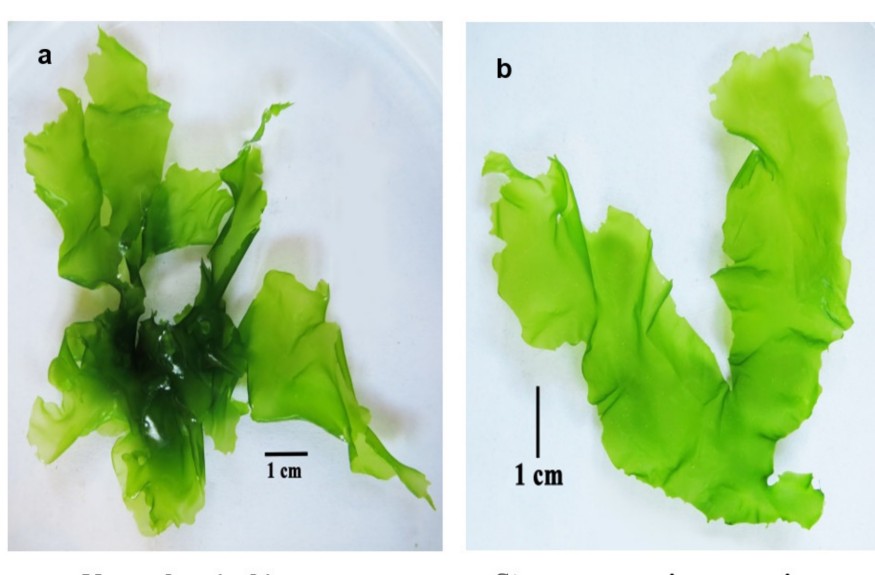

**Figure 3.** Visual (**a**) and stereo microscope (**b**) examination of the collected macroalgae.

The microscopic examination (Figure 4) matched well with what has been previously reported by Ismail and Mohamed [20]. The thallus microscopic examination proved the unordered rounded rectangular to polygonal two-layer cells of *Ulva fasciata*, besides the presence of its characteristic chloroplasts and pyrenoids (1–2/cell). Moreover, the FESEM images (Figure 5a,b) matched well with those previously reported by El-Naggar et al. [21] and El-Sheekh et al. [22]. The obtained EDX analysis results (Figure 5c) also matched well with those reported by El-Naggar et al. [21].

### 3.2. Physicochemical Characterization of the Collected Ulva fasciata

All the listed data in Tables 1–5 are comparable to those recorded for different *Ulva* species [34–45]. The protein content (Table 1) is comparable to that reported by Labib and Hosny [46] for *Ulva fasciata* which recorded 11.68–13.43%, but better than that reported for *Ulva lactuca*, which recorded 7.16% [12]. The collective fiber and carbohydrate content of approximately 54.9% (Table 1) is comparable with that recorded for *Ulva lactuca*, which recorded at 54.04% [12]. The recorded carbohydrate content was higher than that reported for *Ulva fasciata* Delile [47], which recorded 31.5%. Green *Ulva* is reported to have a high ash concentration [44]. The abundant macronutrients in *Ulva* sp. are reported to be calcium, potassium, and sodium regardless of the harvesting season, while the concentrations of micronutrients differ with harvesting month [44]. Usually, Fe and Mn concentrations in *Ulva* sp. are higher than those in brown seaweeds [44]. Green *Ulva* species are usually characterized by a low content of undesirable toxic heavy metals [44]. According to Kumar et al. [43], different green *Ulva* species are usually characterized by high protein, fat, carbohydrate, and crude fiber contents of approximately 6.64–16.38%, 0.5–12%, 17–70.1%, and 29–60.5%, respectively.

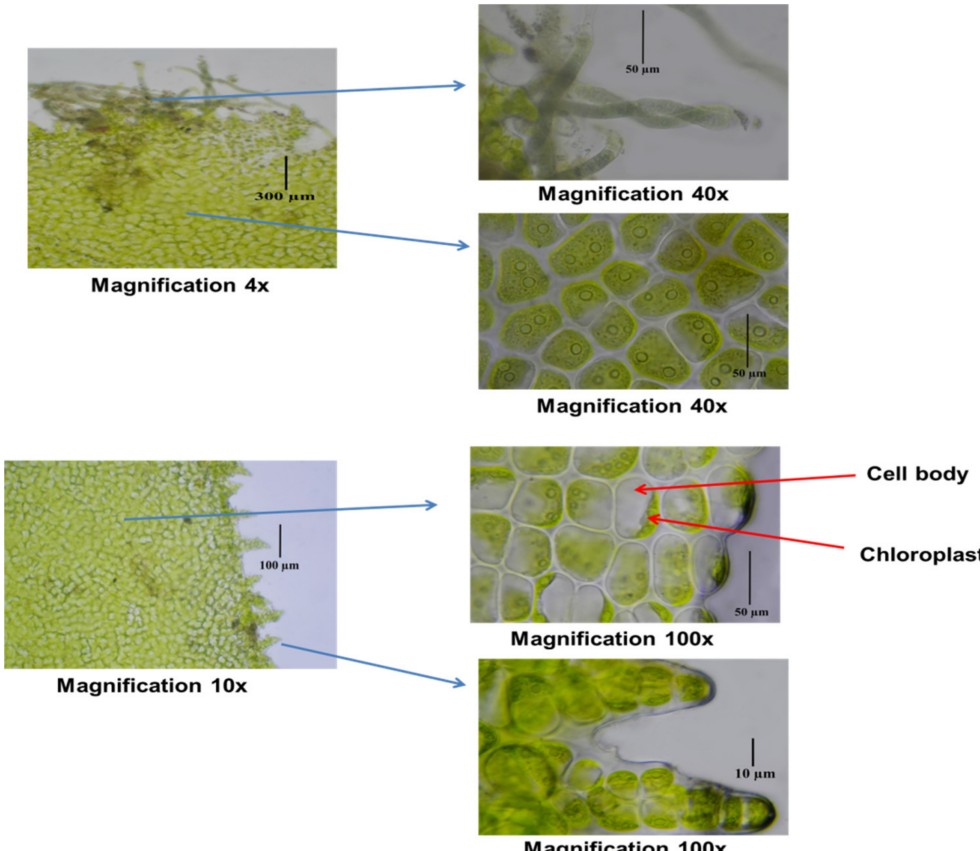

**Figure 4.** Microscopic examination and surface view through middle region of the collected *Ulva fasciata* thallus.

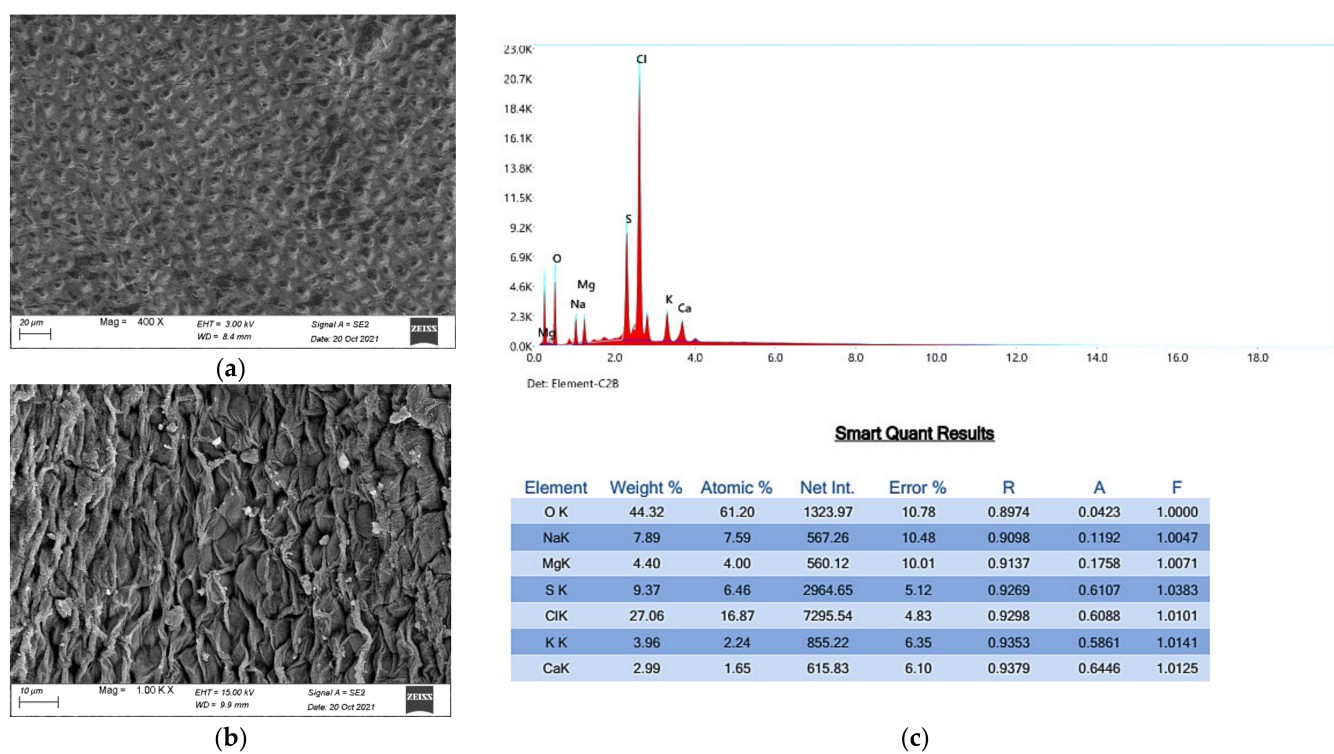

**Figure 5.** The surface-FESEM micrographs (**a**,**b**) and EDX analysis (**c**) of the collected *Ulva fasciata*.

**Table 1.** Nutritional composition of the collected *Ulva fasciata*.

| Dry Matter | Protein Content | Fiber Content | Carbohydrate Content | Lipid Content | Ash Content |
|---|---|---|---|---|---|
| | | **wt% (w:w)** | | | |
| 92.15 | 13.13 | 9.9 | 44.85 | 3.27 | 21 |

**Table 2.** Proximate analysis of the collected *Ulva fasciata*.

| Moisture Content | Volatile Content | Fixed Carbon | Ash Content | Calorific Value MJ/kg |
|---|---|---|---|---|
| 8.5% | 70.7% | 0.3% | 20.5% | 15.19 |

**Table 3.** Cellulose, hemicellulose, and lignin contents of the collected *Ulva fasciata*.

| Hemicellulose | Cellulose | Lignin |
|---|---|---|
| 32.91% | 9.80% | 1.50% |

**Table 4.** Chemical constituents of the collected *Ulva fasciata*.

| Parameter | *Ulva fasciata* | Egyptian Organic Fertilizer Standard (8079/2017) |
|---|---|---|
| Electrical conductivity dS/m | 9.95 | 6–10 |
| Total dissolve solids (mg/L) | 6368 | 3840–6400 |
| Organic C % | 24.59 | Min 15 |
| Organic matter % | 42.30 | Min 18 |
| Moisture content % | 8 | Max 70 |
| Total nitrogen % | 2.1 | Min 0.28 |
| $P_2O_5$ % | 0.755 | Min 0.8 |
| $K_2O$ % | 0.312 | Min 0.8 |
| N + $P_2O_5$ + $K_2O$ % | 3.17 | 4 |
| C/N | 11.71 | 18–22:1 |
| pH | 7.1 | 6–8 |

**Table 5.** Mineral composition of the collected *Ulva fasciata*.

| | Macronutrients | | | | Micronutrients | | | | | Undesired Heavy Metals | | | | | | |
|---|---|---|---|---|---|---|---|---|---|---|---|---|---|---|---|---|
| | P | K | Ca | Mg | Fe | Zn | Mn | Cu | Ni | Cd | Cr | Pb | As | Hg | Co |
| | | mg/kg | | | | mg/kg | | | | | mg/kg | | | | | |
| *Ulva fasciata* | 33 | 260 | 3700 | 82.47 | 180.14 | 66.32 | 15.29 | 3.23 | 0.0519 | 0.591 | 0.31 | 0.0001 | 0.002 | 0.001 | 0.0278 |
| **Egyptian organic fertilizer standard 8079/2017** | — | — | — | — | — | Max 300 | — | Max 100 | Max 180 | Max 5 | Max 300 | Max 300 | — | Max 4 | Max 100 |

The nutritional composition (Table 1) depicts green *Ulva faciata* with considerably high protein, fiber, carbohydrate, and lipid contents, with moderate concentrations of minerals (Table 5), suggesting its applicability in the food industry and as animal feed, besides its medicinal application as a food supplement and as a source of multivitamins. According to the obtained proximate analysis data of the collected *Ulva fasciata* (Table 2) and the

considerable recorded calorific value of 15.19 MJ/kg (Table 2), which is comparable to those derived from lignocellulosic biomass [48], thus it can be used as a solid biofuel. The collected *Ulva faciata* depicted high contents of cellulose and hemicellulose and low lignin content (Table 3), suggesting its application as a feedstock for bioethanol and biobutanol production. Moreover, based on the listed chemical constituents (Table 4) and mineral composition (Table 5) of the collected *Ulva fasciata*, which matched well with the Egyptian standard for solid organic fertilizer, it can be recommended to be used as an ecofriendly organic fertilizer.

### 3.3. Value-Added Products from the Collected Ulva faciata

#### 3.3.1. Pigments and Antioxidants

The UV/Vis spectrum of *Ulva fasciata* ethanol extract (Figure 6) depicts the predominance of the valuable natural chlorophyll$_a$ and chlorophyll$_b$ pigments and the *β*-carotene antioxidant and carotenoid lutein. A similar observation was reported for *Ulva fasciata* Delile [49]. The maximum absorption peaks of chlorophyll$_{a,b}$ appeared at 662 nm and 645 nm, respectively (Figure 6), while, in the blue spectral region between 400 and 500 nm (Figure 6), the carotenoids displayed wide absorption with shoulders. That coincides with those reported for *Ulva fasciata* [37], *Ulva flexuosa* [50], and *Ulva rigida* [51]. Chlorophyll and carotenoids have many applications in the food and pharmaceutical industries.

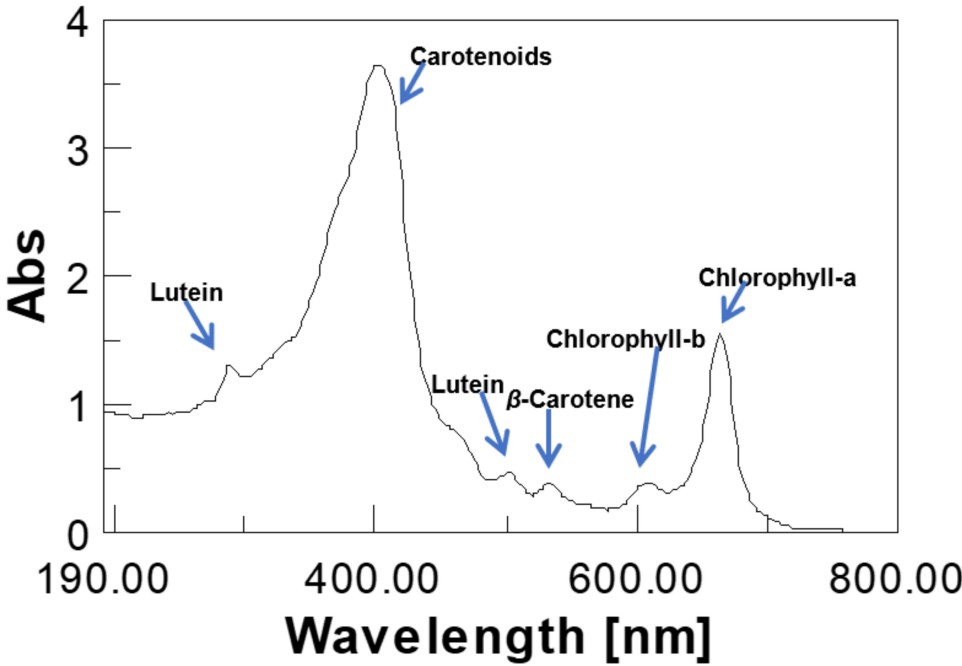

**Figure 6.** UV/Vis spectra of *Ulva fasciata* ethanol extract.

The pigment and anti-oxidant concentrations were calculated according to Fabrowska et al. [50], as follows:

$$Chlorophyll_a \text{ (mg/L)} = (11.75A_{662}) - (2.53A_{645}) \tag{2}$$

$$Chlorophyll_b \text{ (mg/L)} = (18.61A_{645}) - (3.96A_{662}) \tag{3}$$

$$Total\ carotenoids \text{ (mg/L)} = \frac{((1000A_{470}) - (2.27Chloropyll - a) - (81.4Chlorophyll - b))}{227} \tag{4}$$

The algal pigment and anti-oxidant contents were calculated according to Ismail [36] as follows:

$$Chloropyll_a \; (mg/g) = \frac{[12.7A_{663} - 2.69A_{645}] \times V}{W} \tag{5}$$

$$Chloropyll_b \; (mg/g) = \frac{[22.9A_{645} - 4.68A_{663}] \times V}{W} \tag{6}$$

$$Total \; Chloropyll \; (mg/g) = \frac{[20.2A_{645} + 8.02A_{663}] \times V}{W} \tag{7}$$

$$Carotenoids \; (mg/g) = \frac{4A_{480} \times V}{W} \tag{8}$$

That recorded 347.34 mg/L, 14.37 mg/L, and 65.18 mg/L of chlorophyll$_a$, chlorophyll$_b$, and carotenoids, which correspond to 24.59 mg/g, 1.55 mg/g, and 4.05 mg/g, respectively, with total chlorophyll and carotenoids yields of approximately 2.61% and 0.41%, respectively. That is better than those yielded by the previously reported Egyptian *Ulva fasciata* and *Ulva linza*, which recorded 0.219–0.3 mg/g, 0.145–0.268 mg/g, and 0.012–0.105 mg/g chlorophyll$_a$, chlorophyll$_b$, and carotenoids, respectively [52].

### 3.3.2. Mineral-Rich Water Extract

The biochemical analysis of the obtained pale green aqueous extract revealed a conductivity of 13.29 dS/m, total dissolved solids (TDS) of 8502 mg/L, pH 7.7, total organic carbon, organic matter, total nitrogen, $P_2O_5$, and $K_2O$ contents of 2.44%, 4.2%, 0.18%, 0.08%, and 0.29%, respectively, with a C/N of 13.26. The recorded C/N value is nearly the same as that reported for Indian *Ulva fasciata* [8]. The obtained mineral-rich water extract (MRWE) in this study represents approximately 34.89% of *Ulva fasciata's* dry weight. That is better than the previously reported yield for Indian *Ulva fasciata* [8], but lower than that of *Ulva ohnoi* [19], which recorded 26% and 45.42%, respectively. The tabulated mineral composition of *Ulva fasciata* MRWE (Table 6) was comparable to that reported for the Indian *Ulva fasciata* [8]. The MRWE from *Ulva reticulata* [53], *Ulva lactuca* [54], and *Ulva reticulate* [55] has been previously reported for its applicability as a liquid organic fertilizer.

**Table 6.** Mineral compositions of the mineral rich water extract (MRWE).

| | Macronutrients | | | | Micronutrients | | | | | Undesired Heavy Metals | | | | | |
|---|---|---|---|---|---|---|---|---|---|---|---|---|---|---|---|
| | P | K | Ca | Mg | Fe | Zn | Mn | Cu | Ni | Cd | Cr | Pb | As | Hg | Co |
| | mg/L | | | | mg/L | | | | | mg/L | | | | | |
| **MRWE** | 3.49 | 23.78 | 1.34 | 0.54 | 12.08 | 29.62 | 30.25 | 0.571 | 0.003 | 0.312 | 0.003 | 0.001 | 0.0001 | 0.0001 | 0.331 |

### 3.3.3. Starch Yield and Characterization

*Ulva fasciata* yielded starch of approximately 12.55% of its dry weight. That is comparable with the reported extracted starch yields from *Ulva* green seaweeds [13,56] and higher than that reported for *Ulva lactuca* [12] and *Ulva ohnoi* [19], which recorded 9.34% and 3.67%, respectively.

The dynamic light scattering (DLS) analysis of the extracted starch granules (Figure 7a) revealed a bimodal particle size distribution at 158.2 nm and 644.8 nm, due to the occurrence of some aggregates. The DLS analysis depicted starch granules with a major average particle size of approximately 0.2 µm (Figure 7a). That is smaller than that reported for *Ulva ohoni* starch [13]. According to Martins et al. [57], the small sized starch granules up to approximately 2 µm can be used as a fat substitute, as they would be similar to lipid mycelia size.

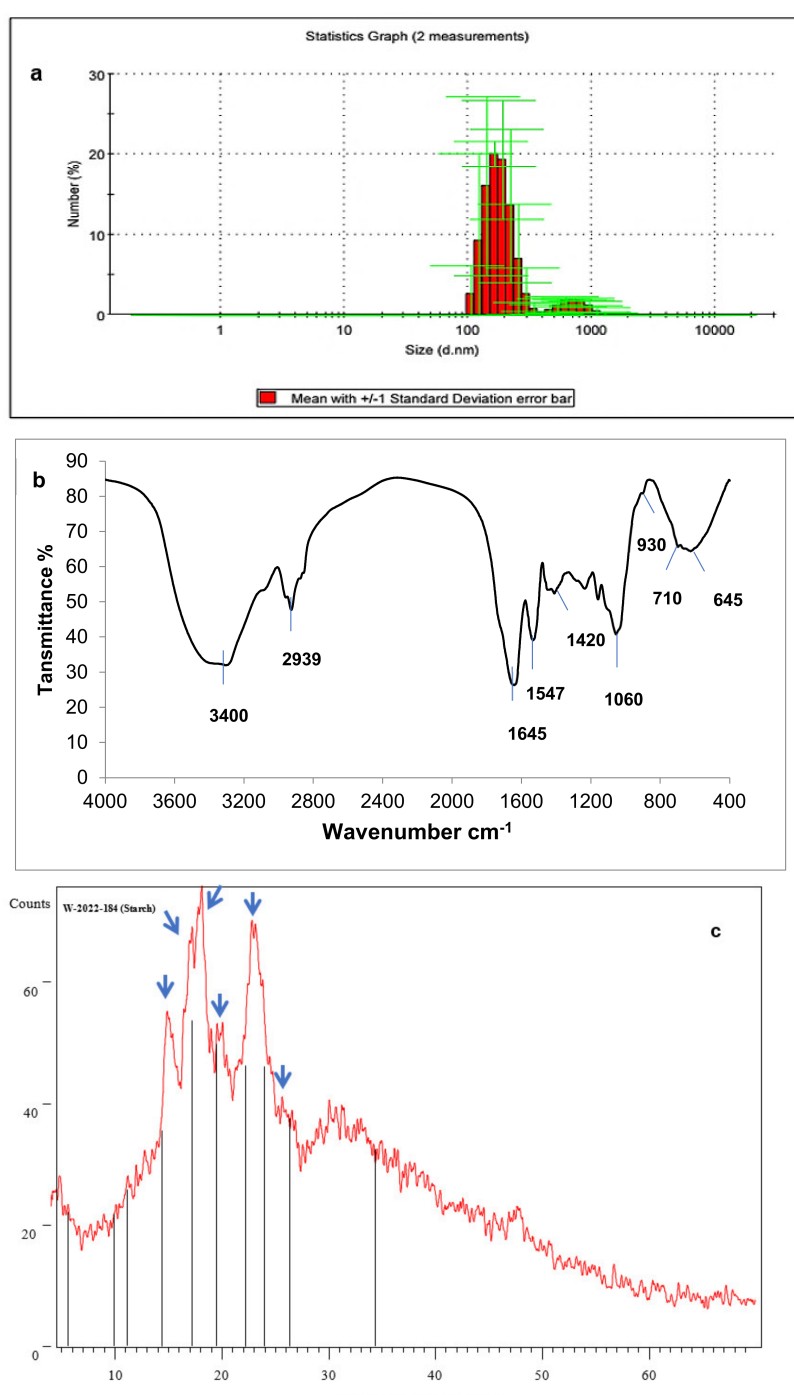

**Figure 7.** Particles size distribution (**a**), FTIR spectrum (**b**), and XRD pattern (**c**) of the extracted *Ulva fasciata* starch.

The Fourier transform infrared (FTIR) analysis (Figure 7b) proved the presence of *Ulva* green seaweed starch with characteristic fingerprint FTIR peaks between 1500 cm$^{-1}$ and 400 cm$^{-1}$ [13,56]. The broad and sharp peaks at 3000–3400 cm$^{-1}$ and 1645 cm$^{-1}$ are denoted for O-H stretching and bending vibrations, respectively. Moreover, the peaks around 2939 cm$^{-1}$ and 1420 cm$^{-1}$ related to the CH$_2$ stretching and bending vibrations, respectively, are also present. The vibrational peaks of pyranoid rings also appeared between 645 and 930 cm$^{-1}$, which coincide with those reported by Karwasra et al. [58]. The vibrational peaks around 930 cm$^{-1}$ are according to Wang et al. [59], denoted for C-O-C of the α-1,4 glycosidic linkage.

The X-ray diffraction (XRD) pattern of the extracted *Ulva fasciata* starch (Figure 7c) proves semi-crystalline polysaccharides; however, the amorphous structure is more prevalent than the crystalline structure. The XRD pattern was found to match well with that of rice starch (card number 00-069-1524 and chemical formula $(C_6H_{10}O_5)_n$). The presence of intense peaks at (2θ) 17°, 18°, and 23° (Figure 7c, blue arrows) and the relatively lower intense peaks at (2θ) 15°, 20°, and 17° (Figure 7c, blue arrows) reveals, according to Wang et al. [59], the starch of an A-type crystalline polymorph structure. Similar observation was reported for *Ulva ohnoi* extracted starch [60]. The observed semi-crystalline XRD pattern is related to the starch polysaccharide structure: linear amylose and branched amylopectin [60,61].

The white spherical and cuboidal starch granules are visible in field emission scanning electron microscope (FESEM) micrographs of the *Ulva fasciata* thallus (Figure 8a, arrows). The extracted starch granules are found to be a mixture of irregular shapes; spherical, pear-shaped, cuboidal, ellipsoidal, and ovoidal, but all are characterized by a smoothed surface structure (Figure 8b). Similar observations were reported by Prabhu et al. [13,56]. The energy dispersive X-ray spectroscopy (EDX) analysis (Figure 8c) depicted the purity of the extracted starch as being mainly composed of 28.32% carbon and 60.52% oxygen, with a minor content of sodium and magnesium, recording 3.14% and 1.45%, respectively. It has also a nitrogen content of 3.95%, revealing a protein content of approximately 24.69%, as calculated according to Alves et al. [7]. The protein content of the extracted starch in this study is higher than that reported for *Ulva ohoni*, which has 0.08% protein content [13]. The EDX analysis (Figure 8c) showed traces of sulfur, potassium, calcium, and iron, recording 0.52%, 0.29%, 0.22%, and 0.41%, respectively.

The thermogravimetric analysis (TGA) curve (Figure 9a) matches that reported for *Ulva ohnoi* extracted starch [13]. Under non-oxidative degradation, a mass loss of 5.96% from 39 °C to 100 °C was recorded, which would be related to the loss of residual moisture. From 100 °C to 313 °C the weight loss recorded was approximately 20.43%, which might be attributed to the volatile constituents of starch. The weight loss between 313 °C and 600 °C recorded 22.3%, while between 600 °C and 900 °C recorded 29.75%. That would be related to the decomposition of organic matter and the carbonization of the starch, organic and non-volatile constituents, respectively.

The deferential weight loss curve, i.e., the differential thermogravimetry (DTG) curve (Figure 9b) displayed three main decomposition peaks at 49 °C, 236 °C, and 285 °C. According to Kumar et al. [62], starch granules degraded during thermal analysis due to dehydration, glycogen production, and depolymerization. The extracted *Ulva fasciata* starch displayed four endothermic peaks at 59 °C, 295 °C, 312 °C, and 712 °C, respectively, on the differential thermal analysis (DTA) curve (Figure 9c), which supported the TGA findings (Figure 9a). It can be depicted from the DTA curve (Figure 9c) that the melting phase transition point of the extracted *Ulva fasciata* starch from a solid, crystalline state to an amorphous molten state occurs at 59 °C, which coincides with Prabhu et al. [13] findings for *Ulva ohoni* starch, which recorded 58.4 °C. However, the main decomposition actually starts at 236 °C, indicating sufficient thermal stability. It can also be depicted from the differential scanning calorimetry (DSC) curve (Figure 9d) that the gelatinization temperature of the thermally induced hydration and swelling of the starch granules occurred between 50–70 °C at 56.52 °C. Moreover, an endothermic peak was observed with onset ($T_o$ 79.86 °C), peak (Tp 119.65 °C), and conclusion ($T_c$ 162.47 °C) temperatures (Figure 9d). The gelatinization enthalpy (ΔH), which is the required energy for breaking the intermolecular bonds during gelatinization was recorded at 24.29 J/g. That agrees with the data obtained for *Ulva ohoni* starch [13] and indicates the relatively high amylose content, crystalline nature, and large sized starch granules. The obtained Tp 119.65 °C value was relatively similar to that of *Ulva ohoni starch*, $T_p$ 118.4 °C [13], but higher than that of Floridean starch from red algae *Gracilariopsis lemaneiformis* and *G. chilensis*, which recorded $T_p$ 55.1 °C and 52.7 °C, respectively.

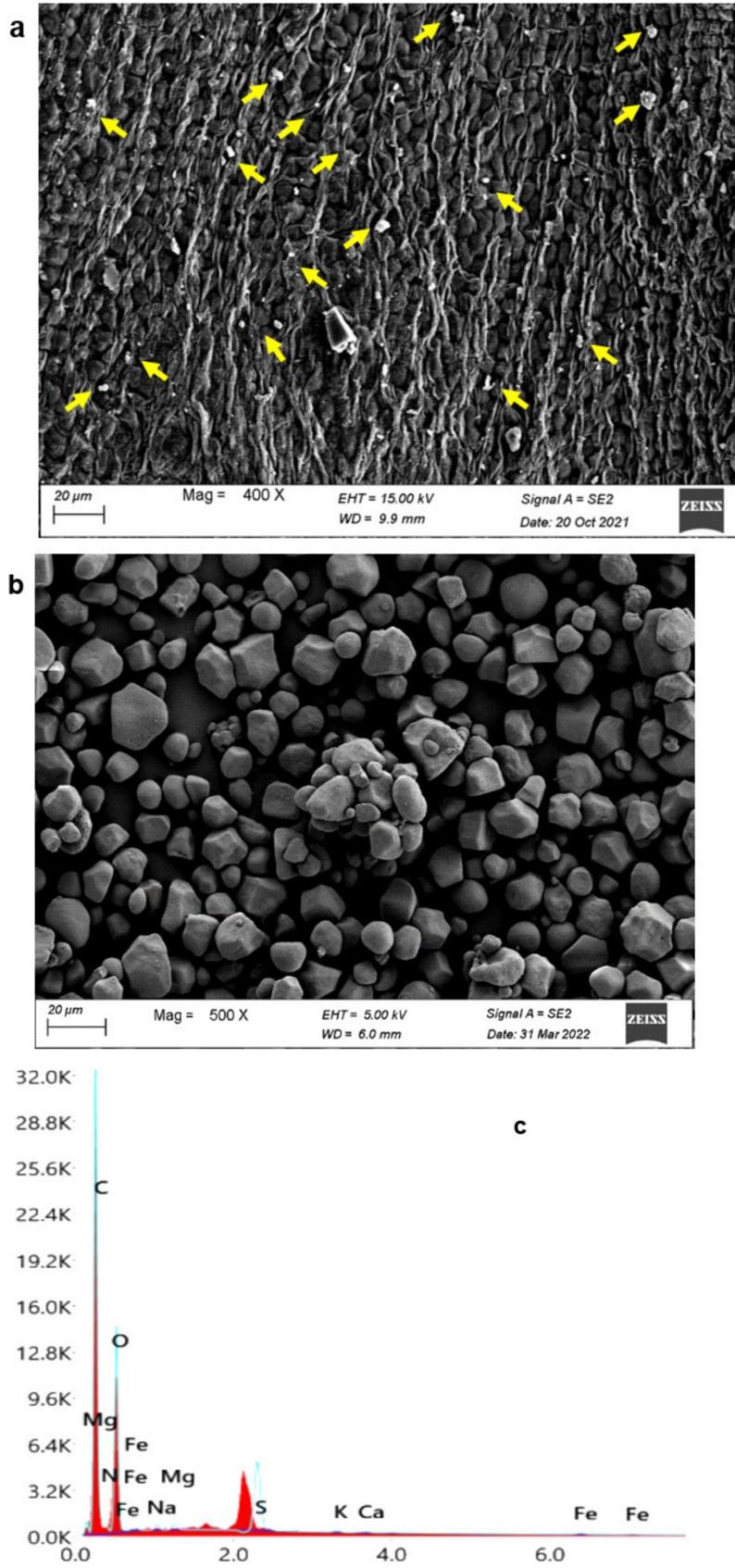

**Figure 8.** FESEM images of *Ulva fasciata* thallus showing the starch granules – yellow arrows (**a**), its extracted starch granules (**b**), and EDX of extracted starch granules (**c**).

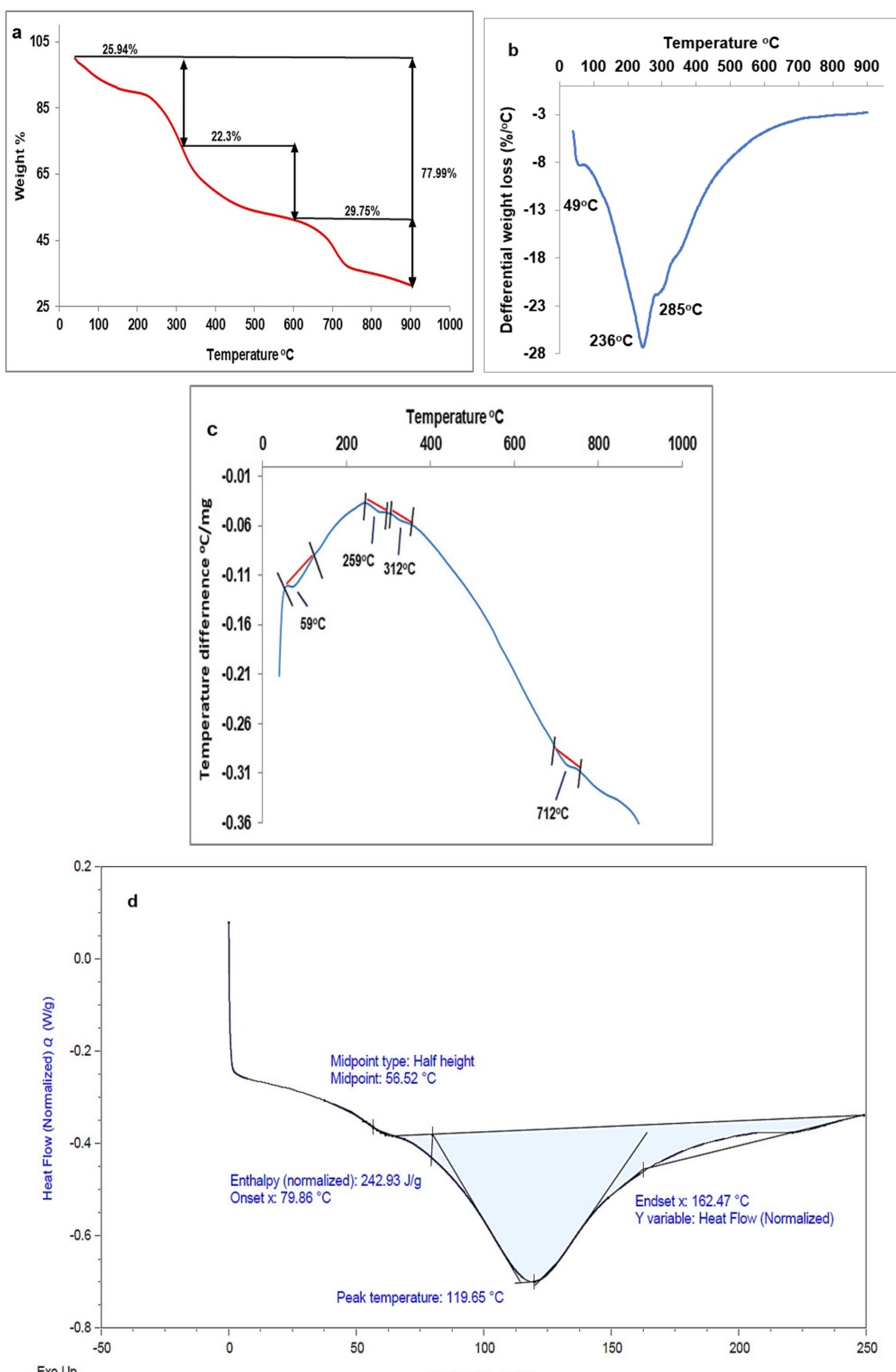

**Figure 9.** The TGA (**a**), DTG (**b**), DTA (**c**), and DSC (**d**) curves of the extracted *Ulva fasciata* starch.

### 3.3.4. Lipid Yield and Profile

*Ulva fasciata* yielded lipids of approximately 3.27% of its dry weight. That is better than those recorded for Indian *Ulva fasciata*, *Ulva lcatuca*, *Ulva rigida*, Ghanaian *Ulva faciata*, and other *Ulva* sp., which recorded 2.7% [8], 2.36% [12], 2.53% [63], 2.6% [2], and 1–1.3% [44], respectively, and comparable with that reported for *Ulva ohnoi* [19], which recorded 3.81%.

The summarized fatty acid profile of the extracted lipid (Table 7) matches that reported for India *Ulva fasciata* [8]. The predominance of saturated fatty acids relative to mono- and poly-unsaturated fatty acids was also reported for *Ulva rigida* [63] and *Ulva* sp. [44]. The Egyptian *Ulva fasciata* in this study has a high concentration of palmitic acid (C16:0) recording approximately 38.93%; this agrees well with that reported by Trivedi et al. [8] and Samarasinghe et al. [44] for Indian *Ulva fasciata* and *Ulva* sp., which recorded 37.4–39.8% and 36.8–38.1%, respectively. The low recorded n-3/n-6 ratio (0.13) emphasizes the potential benefits of consuming this macroalga for human health. The recorded high concentrations of omega-3 fatty acids (n-3 PUFAs) and omega-9 fatty acids (n-9 MUFAs) with relatively low omega-6 fatty acids (n-6 PUFAs) are beneficial as n-3 PUFAs and n-9 MUFAs have anti-inflammatory, antioxidant, and anti-cancer properties, and they improve cardiovascular performance, while n-6 PUFAs enhance tumor growth, inflammation, and depression symptoms [63].

**Table 7.** Fatty acid composition of the collected *Ulva fasciata*.

| Fatty Acid | Trivial Name | Content |
| --- | --- | --- |
| C8:0 | Caprylic acid | 1.17% |
| C10:0 | Capric acid | 0.84% |
| C14:0 | Myristic acid | 4.45% |
| C16:0 | Palimitic acid | 38.93% |
| C17:0 | Margaric acid | 1.63% |
| C18:0 | Stearic acid | 3.72% |
| C20:0 | Arachidic acid | 0.39% |
| C22:0 | Behenic acid | 2.64% |
| C24:0 | Lignoceric acid | 0.92% |
| C16:1 (n-9) | Palmitoleic acid | 3.08% |
| C17:1 | Heptadecenoic acid | 1.84% |
| C18:1 (n-9) | Oleic acid | 1.16% |
| C18:2 (n-6) | Linoleic acid | 1.42% |
| C18:3 (n-3) | α-Linolenic acid | 15.68% |
| C18:3 (n-6) | γ-Linolenic acid | 2.95% |
| C18:4 (n-3) | Stearidonic acid | 16.15% |
| C20:1 (n-9) | Gondoic acid | 1.16% |
| C20:5 (n-3) | Eicosapentaenoic acid | 1.87% |
| Total saturated fatty acids | | 54.69% |
| Total mono-unsaturated fatty acids (MUFAs) | | 7.24% |
| Total poly-unsaturated fatty acids (PUFAs) | | 38.07% |
| Total (n-3) | | 33.70% |
| Total (n-6) | | 4.37% |
| Total (n-9) | | 5.4% |
| n-6/n-3 | | 0.13 |

### 3.3.5. Ulvan Yield and Characterization

The collected *Ulva fasciata* yielded approximately 22.24% purified ulvan. That is comparable with the previously reported extracted ulvan yields from *Ulva rotundata* [4], *Ulva rigida* [64], *Ulva lactuca* [6], and Indian *Ulva fasciata* [8], which recorded 21.5%, 24.3%, 21.68–32.67%, and 25%, respectively. However, it was higher than that reported for *Ulva ohnoi* [19] which recorded 13.88%. The DLS analysis depicted ulvan with an average granule size of approximately 1.58 μm (Figure 10a).

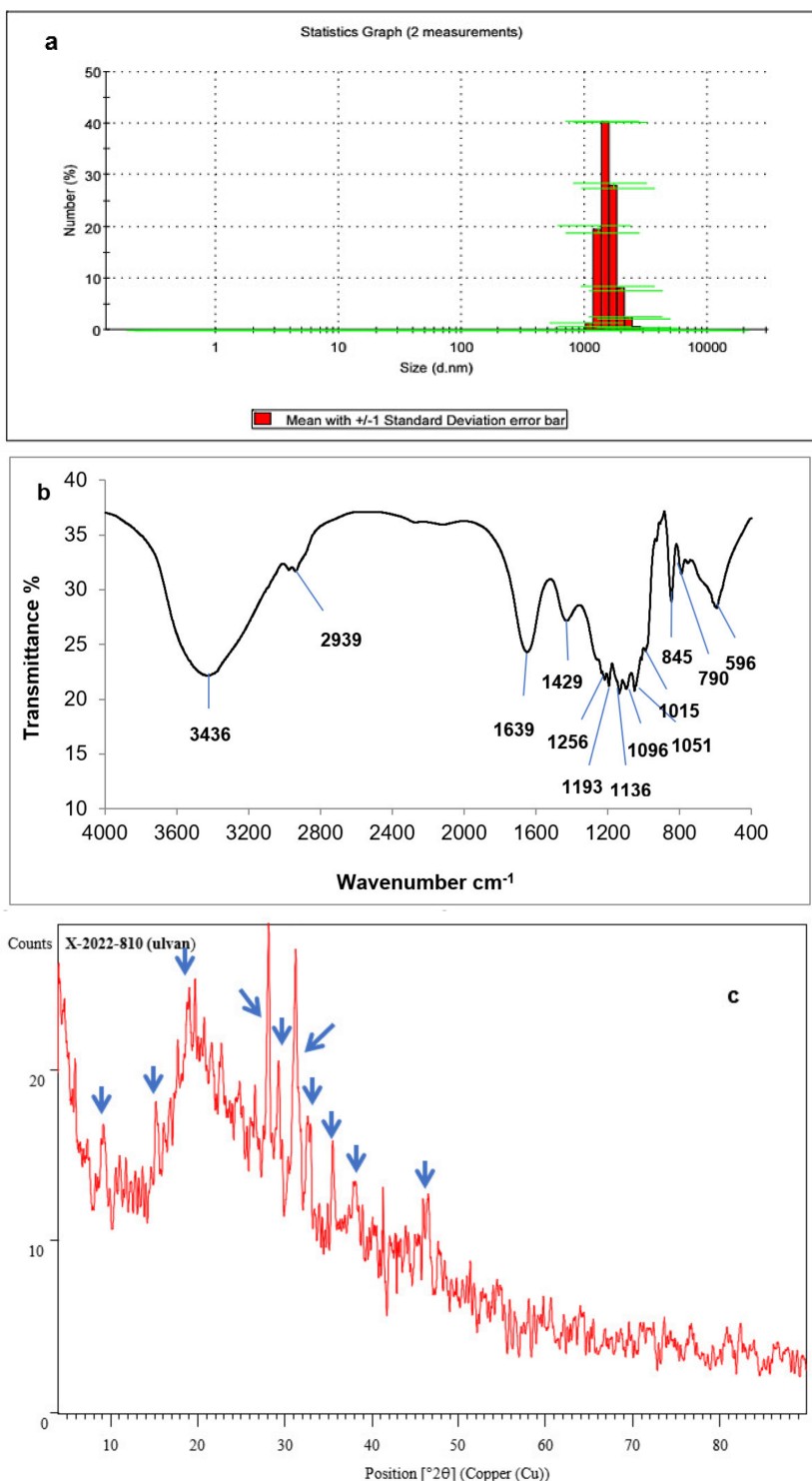

**Figure 10.** Particles size distribution (**a**), FTIR spectrum (**b**), and XRD pattern (**c**) of the extracted *Ulva fasciata* ulvan.

The FTIR spectrum of the extracted ulvan (Figure 10b) confirmed its identity and purity. According to Yaich et al. [65], the absorption peaks in the region between 1256 cm$^{-1}$ and 1051 cm$^{-1}$ are considered the fingerprint for ulvan. It is similar to that extracted from *Ulva rigida* [66], *Ulva pertusa* [67], and Indian *Ulva fasciata* [8]. The characteristic band at 3436 cm$^{-1}$ (Figure 10b) is ascribed to -OH groups and consistes with the structure of disaccharides in Ulvan. According to Trivedi et al. [8], the FTIR bands at 1256 and 845 cm$^{-1}$ (Figure 10b) are indicative of the S=O and C-O-S stretching of the sulfate groups, respectively. According to Pengzhan et al. [67] the two vibrational bands at 845 cm$^{-1}$ and 790 cm$^{-1}$ (Figure 10b) indicate the existence of sulfate ester substitutions. While the bands 1639 cm$^{-1}$ and 1429 cm$^{-1}$ (Figure 10b) are, according to Aguilar-Briseño et al. [68], allocated to the symmetric and asymmetric vibrations of C=O groups of the uronic acid in Ulvan product. Moreover, the peaks at 1096 cm$^{-1}$ and 1051 cm$^{-1}$ (Figure 10b), relate to the C–O–C stretching of glucosides [67]. Nevertheless, the peaks at 1193 cm$^{-1}$, 1136 cm$^{-1}$, and 1015 cm$^{-1}$ (Figure 10b) are, according to Bayro et al. [69], indicative of the presence of pyranose rings, i.e., the six-membered ring saccharides. The other absorption band at 2939 cm$^{-1}$ (Figure 10b) is, according to Bayro et al. [69], associated with the stretching signals of C-H groups in the configuration of sugar polysaccharides in the structure of ulvan.

The XRD pattern of the extracted *Ulva fasciata* ulvan (Figure 10c) proves semi-crystalline polysaccharides, with two major peaks at (2θ) 28.15° and 31.22° (Figure 10c, blue arrows), and other minor peaks at (2θ) 16°, 19.83°, 29.66°, 38.98°, and 49.20° (Figure 10c, blue arrows). A similar observation was reported for *Ulva* sp. ulvan [7], Indian *Ulva fasciata* ulvan [70], and Egyptian *Ulva fasciata* ulvan [11]. The minor peaks appear at (2θ) 9°, 33°, and 36° (Figure 10c, blue arrows), were also reported for Egyptian *Ulva fasciata* ulvan [10]. However, it is clear from the XRD pattern (Figure 10c) that the amorphous structure is more prevalent than the crystalline structure. Similar observation was previously reported by Madany et al. [11], who attributed that to the heavily branched structure of ulvan because of its indistinct backbone. Similar observation was also reported by Gajaria et al. [70] and attributed that to the presence of junction zones, where cross-linking occurs. Moreover, Madany et al. [11] reported that the ulvan's disordered conformational structure, which is caused by its heterogeneous chemical composition, accounts for the presence of amorphous regions. However, the repeated aldobiouronic units could make its crystalline region more distinct.

The FESEM examination proved an amorphous structure and rough textured ulvan biopolymer (Figure 11a) with a porous cross-linked scaffold surface that seems to have a three-dimensional network structure (Figure 11b). The FESM micrograph at higher magnification power (25,000×, Figure 11c) reveals spherical particles with a size range of 20–40 nm, some are dispersed, some other are cross-linked, and some are in aggregates covered with a thin layer of organic matrix. Similar observation was reported for *Ulva lactuca* ulvan [71]. The EDX analysis (Figure 11d) revealed that the major components of the produced ulvan are carbon, oxygen, and sulfur, which compromised 16.64, 34.47, and 22.18%, respectively. That agreed with the ulvan extracted from the Egyptian *Ulva lactuca* [71]. The EDX analysis (Figure 11d) also revealed that the extracted ulvan was composed of 3.05, 8.35, 3.19, 8.46, 3.17, and 0.5% nitrogen, sodium, magnesium, potassium, calcium, and iron, respectively. Thus, the EDX analysis (Figure 11d) confirmed a sulfated polysaccharide with considerable protein content. The C-content in the extracted ulvan from the Egyptian *Ulva fasciata* in this study is lower than that reported for Indian *Ulva fasciata* [70], while the S content is higher. The protein content was calculated according to Alves et al. [7] by multiplying the N-content by the factor 6.25, which revealed a protein content of approximately 19.06%. That is higher than the reported protein content of *Ulva clathrata* ulvan [5] and *Ulva armoricana* ulvan [72], but within the range of *Ulva rotundata* ulvan [4]. The presence of Na, K, Ca, Mg, and Fe was also reported in *Gracilaria gracilis* and *Ulva lactuca* sulfated polysaccharide [73] and *Ulva lactuca* ulvan [71].

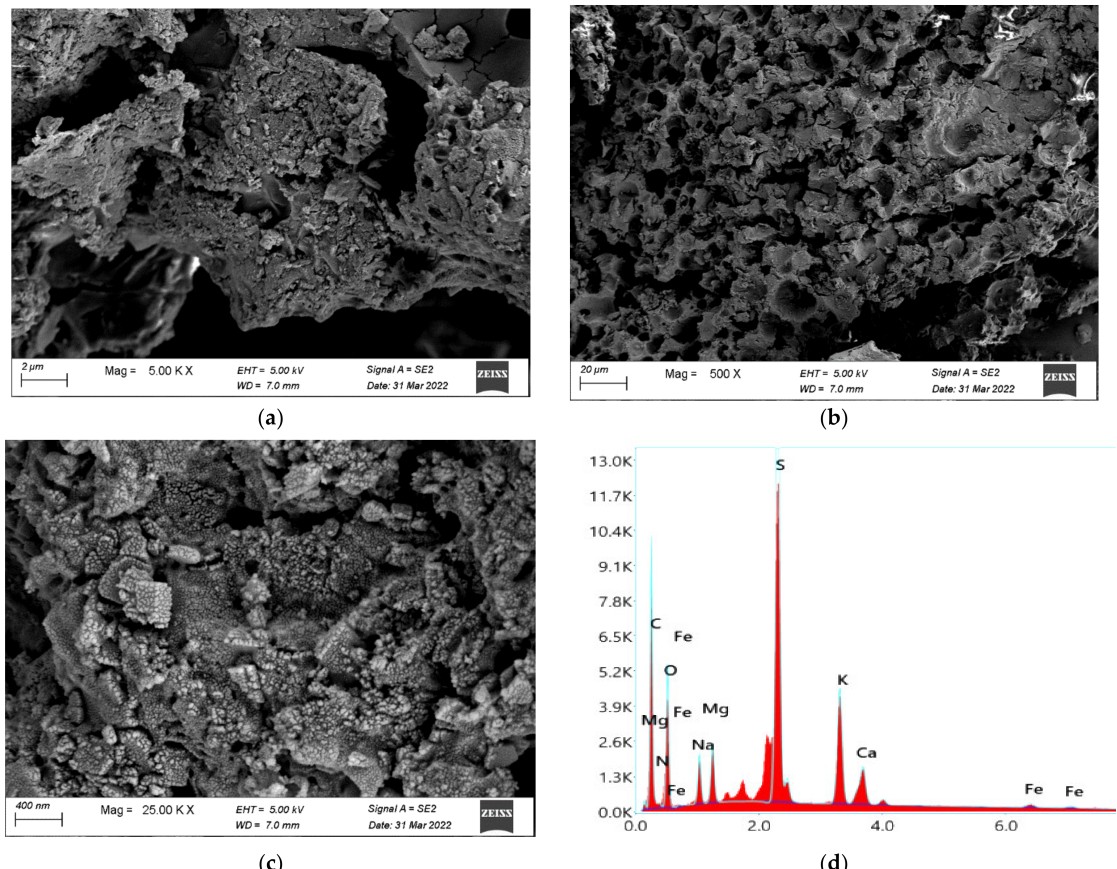

**Figure 11.** FESEM images (**a**–**c**) and EDX (**d**) of the extracted *Ulva fasciata* ulvan granules.

The TGA curve (Figure 12a) matches that reported for ulvan extracted from *Ulva* sp. [7] and *Ulva lactuca* [65]. Under non-oxidative degradation, a mass loss of 2.53% from 39 °C to 100 °C was recorded, which would be related to the loss of residual moisture. From 100 °C to 332 °C the weight loss was approximately 23.94%, which might be attributed to volatile constituents of ulvan. The weight loss between 332 °C and 600 °C recorded 25.06%, while between 600 °C and 900 °C recorded 15.43%. That would be related to the decomposition of organic matter and the carbonization of the ulvan organic and non-volatile constituents, respectively. The ulvan ash content recorded approximately 32.59%. The DTG curve (Figure 12b) reveals one major decomposition peak at 220 °C, where ulvan biopolymer starts to depolymerize and the degradation of monomers occurs. This significant amount of ash might be attributed to the counterions linked to sulphate groups and uronic acids in ulvan, which also match with the aforementioned results of EDX analysis.

The extracted *Ulva fasciata* ulvan displayed two main endothermic peaks at 327 °C, and 730 °C, respectively, on the DTA curve (Figure 12c), revealing two main thermal decomposition stages and confirming the two main decomposition stages in the TGA curve (Figure 12a). Similar observation was reported for *Ulva lactuca* ulvan [65]. Thus, from TGA and DTA results, it can be depicted that the extracted *Ulva fasciata* ulvan in this study is highly thermally stable, tolerating up to approximately 220 °C without a noticeable thermal decomposition. That is better than reported for ulvan extracted from *Ulva* sp. [7] and *Ulva lactuca* [65], which depicted thermal stability up to 200 °C and 180 °C, respectively. The DSC thermogram (Figure 12d) represents two main endothermic regions and one main exothermic region. Similar observation was reported for *Ulva fasciata* ulvan [11] and *Ulva lactuca* Linn ulvan [74]. The DSC thermogram depicts a low glass transition temperature ($T_g$) at around 5 °C and a major transition temperature at about 120 °C (Figure 12d). Similar observation was reported by Alves et al. [7]. The melting temperature ($T_m$) peaked at 227.91 °C (Figure 12d), and coincides with that reported for *Ulva lactuca* Linn ulvan [74].

According to Arul Manikandan and Lens [75], the higher melting point of ulvan indicates a close compatibility with petroleum-based polymers and recommends its application in the packaging industry.

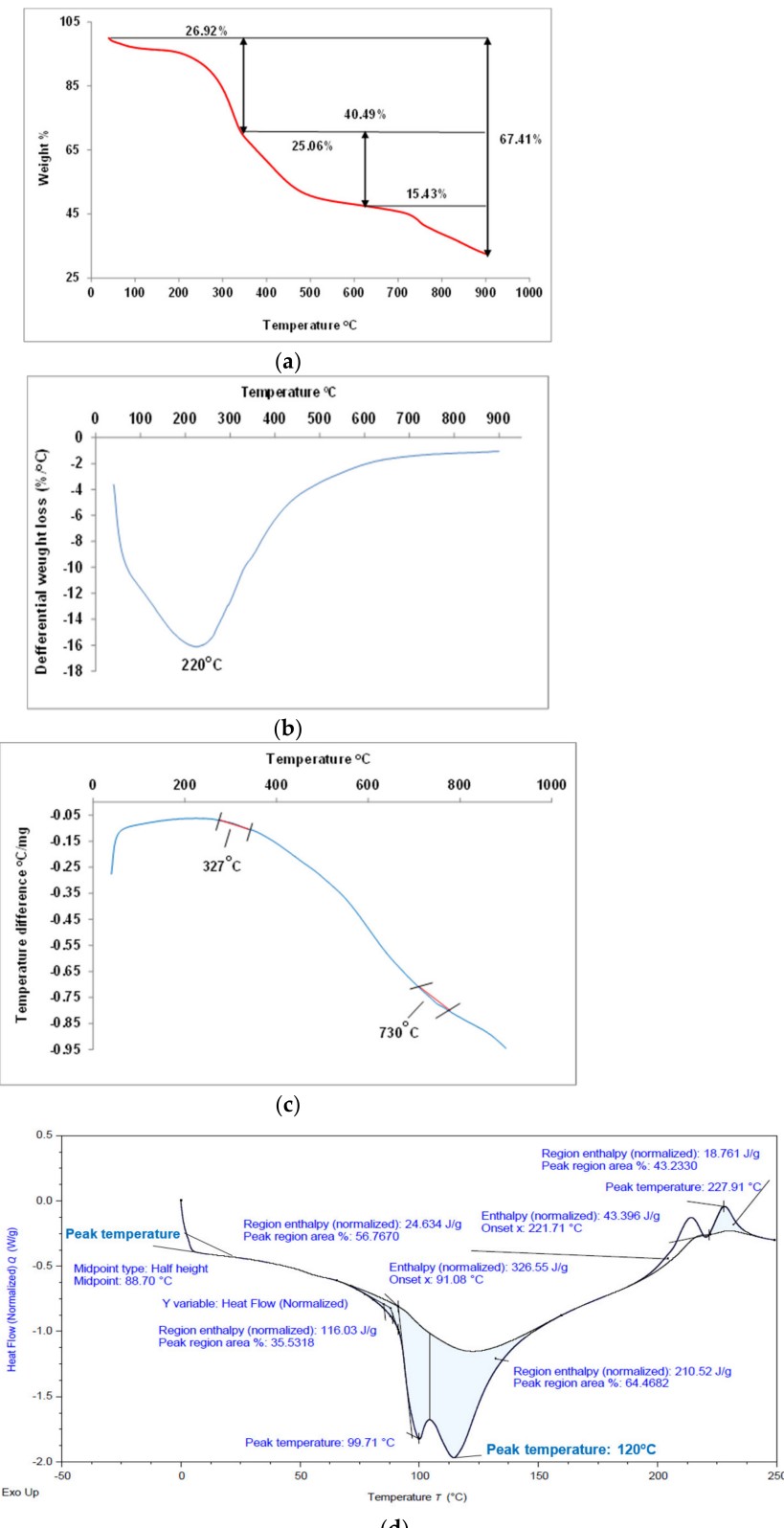

**Figure 12.** The TGA (**a**), DTG (**b**), DTA (**c**), and DSC (**d**) curves of the extracted *Ulva fasciata* ulvan.

### 3.3.6. The Antimicrobial Activity of the Extracted *Ulva fasciata* Ulvan

It can be depicted from the data listed in Table 8 and illustrated in Figure 13 that the extracted *Ulva fasciata* ulvan showed efficient biological activity against the tested pathogenic Gram-positive *B. subtilis* and *S. aureus* and Gram-negative *E. coli* and *P. putida* and yeast *C. albicans*, however, it expressed a relatively sufficient antifungal activity against the tested *A. niger*. Nevertheless, the antimicrobial activity of the extracted *Ulva fasciata* ulvan in this study was better than that reported for *Ulva reticulate* ulvan [76] and *Ulva lactuca* ulvan [71]. The extracted *Ulva fasciata* ulvan showed a comparable inhibitory effect relative to the standard antibiotics except for the tested fungal strain, recording an activity index of approximately $0.89 \pm 0.045$, $0.91 \pm 0.046$, $0.86 \pm 0.043$, $0.94 \pm 0.047$, $0.77 \pm 0.039$, and $0.59 \pm 0.03$ against *B. subtilis* ATCC 6633, *S. aureus* ATCC 35556, *E. coli* ATCC 23282, *P. putida* ATCC 10145, *C. albicans* IMRU 3669, and *A. niger* ATCC 16404, respectively. Ibrahim et al. [71] mentioned that the *Ulva lactuca* ulvan does not express any anti-fungal effect against *A. niger* or antibacterial activity against the Gram-positive *S. aureus*. The MIC and MLC recorded $312.5 \pm 2.5$ mg/L for the Gram-positive *B. subtilis* ATCC 6633 and *S. aureus* ATCC 35556, the Gram-negative *P. putida* ATCC 10145, the yeast *C. albicans* IMRU 3669, and the fungi *A. niger* ATCC 16404, while $625 \pm 3.5$ mg/L for the Gram-negative *E. coli* ATCC 23282. The efficient antimicrobial activity of the extracted *Ulva fasciata* ulvan (Table 8 and Figure 13) suggests its application for the preparation of wound dressing material and water disinfection from pathogenic microorganisms. Ulvan also showed efficient biocidal activity against halophilic SRB (Table 9) with a MLC of 1000 mg/L against both the standard non-halotolerant *Desulfovibrio sapovorans* ATCC 33892 and the collected mixed halotolerant mixed culture.

**Table 8.** The antimicrobial activity of the extracted *Ulva fasciata* ulvan relative to standard antibiotics *.

| Tested Microorganisms Compound ID | *B. subtilis* ATCC 6633 | *S. aureus* ATCC 35556 | *E. coli* ATCC 23282 | *P. putida* ATCC 10145 | *C. albicans* IMRU 3669 | *A. niger* ATCC 16404 |
|---|---|---|---|---|---|---|
| Ulvan | $31 \pm 0.62$ | $32 \pm 0.64$ | $19 \pm 0.38$ | $31 \pm 0.62$ | $20 \pm 0.4$ | $13 \pm 0.26$ |
| Reference antibiotic | $35 \pm 0.7$ | $35 \pm 0.7$ | $22 \pm 0.44$ | $33 \pm 0.66$ | $26 \pm 0.52$ | $22 \pm 0.44$ |

* Expressed in mm.

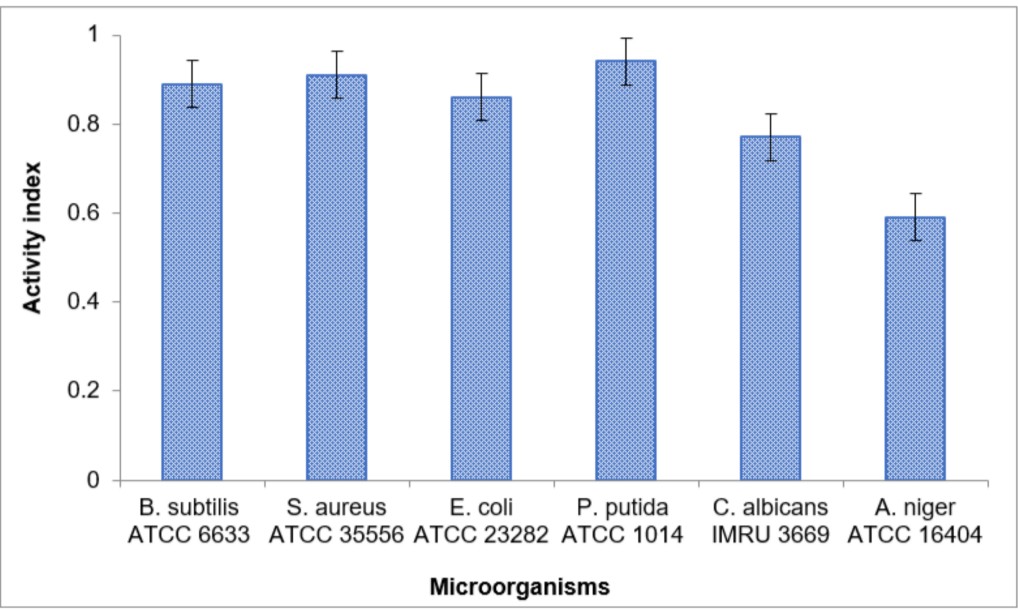

**Figure 13.** Activity index of the extracted *Ulva fasciata* ulvan.

**Table 9.** The biocidal activity of extracted ulvan against halophilic planktonic sulfate reducing bacteria (SRB).

| Tested SRB / Sample ID | *Desulfovibrio sapovorans* ATCC 33892 (Cell/mL) | | | | Mixed Culture (Cell/mL) | | | |
|---|---|---|---|---|---|---|---|---|
| +ve control | $>3 \times 10^7$ | | | | $>3 \times 10^7$ | | | |
| −ve control | Nil | | | | Nil | | | |
| Ulvan concentration mg/L | 500 | 1000 | 200 | 3000 | 500 | 1000 | 2000 | 3000 |
| | $1.5 \times 10^3$ | Nil | Nil | Nil | $2.6 \times 10^4$ | Nil | Nil | Nil |

The recorded ulvan biocidal activity against SRB (Table 9) is much better than those reported for the hot-water extract of both mandarin and orange peels, which recorded MLC of 2500 mg/L and 10,000 mg/L for non-halotolerant *Desulfovibrio sapovorans* ATCC 33892 and planktonic halotolerant mixed SRB culture, respectively [40,77]. The obtained results were also better than those reported for hot water extracts of both onion and garlic peels, which did not express any biocidal activity against the non-halotolerant *Desulfovibrio sapovorans* ATCC 33892 up to 3000 mg/L [77]. To our knowledge, this is the first time to record ulvan biocidal activity against SRB, and it can be recommended as a green biocide instead of the expensive, toxic, and non-ecofriendly chemical biocides for mitigation of microbially influenced corrosion in the oil and gas industry.

### 3.3.7. Protein Yield

*Ulva fasciata* yielded approximately 13.37% crude protein. That is comparable with the previously reported extracted protein yields from *Ulva ohnoi* [19], *Ulva* species [44], but higher than that reported for *U. rigida* [38] which recorded 14.83%, 13.3–15.9%, and 8.5%, respectively.

### 3.3.8. Cellulose Yield and Characterization

*Ulva fasciata* yielded approximately 10.66% cellulose powder. That is comparable with the previously reported extracted cellulose yield from Indian *Ulva fasciata* [8] and Egyptian *Ulva fasciata* [52], but higher than that extracted from *Ulva ohnoi* [19], *Ulva lactuca* [34], and *Ulva linza* [52], which recorded 10%, 10.04%, 2.2%, and 8.7%, respectively. The extracted cellulose was 85% pure, with about 2% hemicellulose and 0.06% lignin. The purity of the extracted cellulose is about the same as what Prabhu et al. [19] reported, which was 80.19%.

The DLS analysis of the extracted *Ulva fasciata* cellulose showed a bimodal particle size distribution (Figure 14a) at 90.82 nm and 420.2 nm, with a major average particle size in the nanoscale, recording approximately 91 nm (Figure 14a).

The FTIR spectrum of the extracted *Ulva fasciata* cellulose (Figure 14b) shows the peaks of polysaccharides and matches well with that reported for Indian *Ulva faciata* [8] and Egyptian *Ulva lactuca* [15]. The broad and sharp peaks around 3414 cm$^{-1}$ and 1636 cm$^{-1}$ (Figure 14b) are, according to Jmel et al. [78] denoted for O-H stretching and bending vibrations, respectively, while the peak around 2922 cm$^{-1}$ is consistent with the CH$_2$ stretching vibration of cellulose. The peaks that appear between 1020 cm$^{-1}$ and 1050 cm$^{-1}$, besides that appears at 1060 cm$^{-1}$ (Figure 14b) correspond, according to Salem and Ismail [52], to the symmetric stretching and bending vibrations of C–O–C, respectively. The appeared peaks at 1420 cm$^{-1}$ and 1316 cm$^{-1}$ are, according to Wahlström et al. [34] related to the H–C–H scissor and H–C–H tip vibrations of α-cellulose. The characteristic small FTIR-vibration peak of the β-glycosidic bond, which is an indicator for the presence of the cellulose polymeric polysaccharides [78] appears around 890 cm$^{-1}$ (Figure 14b), which is also confirmed by the presence of carbohydrates C–OH out-of-plane bending vibrations at 618 cm$^{-1}$ and 669 cm$^{-1}$. The peak at 1112 cm$^{-1}$ would according to Gomaa et al. [15] indicate the presence of β-glycosidic linkages between the cellulose anhydroglucose rings.

The absence of ulvan sulfated polysaccharide with characteristic S=O and C–O–S stretching peaks around 1220 cm$^{-1}$ and 840 cm$^{-1}$ would confirm the efficient extraction of ulvan in the previous step. The peak at 1542 cm$^{-1}$ (Figure 14b), which corresponds to N-H bending vibration (Figure 14b) would indicate the presence of protein in the extracted starch fraction [34]. Additionally, the presence of a peak around 1710 cm$^{-1}$ of C=O (Figure 14b) would, according to Wahlström et al. [34], indicate to some extent the possible existence of xyloglucan and cellulose oxidation during the bleaching step. Nevertheless, the absence of a shoulder peak at around 1730 cm$^{-1}$ (Figure 14b) would imply that neither lignin's ester linkage nor the acetyl and uronic ester groups of the hemicelluloses and ulvan exist [78]. Besides, the absence of peaks at 1413 cm$^{-1}$ and 1533 cm$^{-1}$ also denotes the absence of the aromatic ring's C=C in lignin [9].

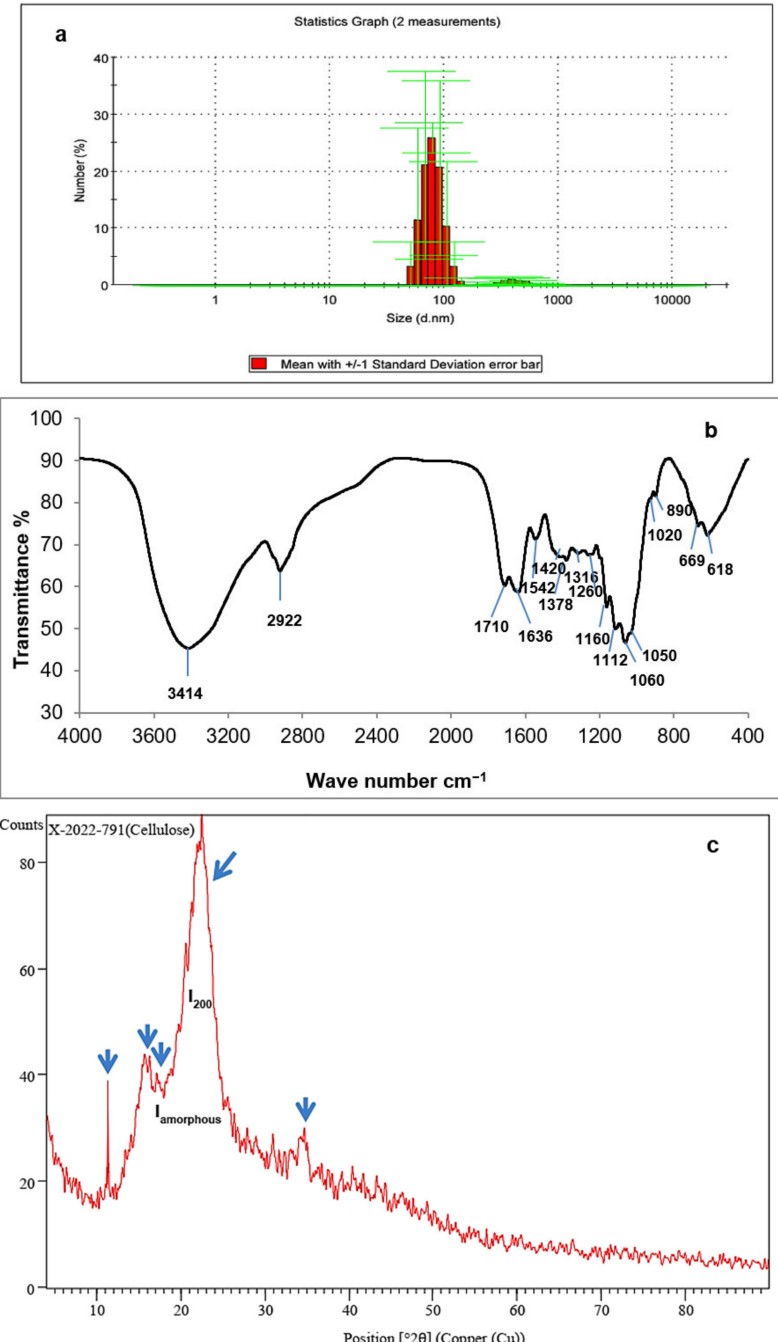

**Figure 14.** Particles size distribution (**a**), FTIR spectrum (**b**), and XRD pattern (**c**) of the extracted *Ulva fasciata* cellulose nanofibrils.

The XRD pattern of the extracted *Ulva fasciata* cellulose (Figure 14c) was found to be well matched with that of cellulose (card number 00-003-0192 and chemical formula $(C_6H_{10}O_5)$) and agrees with that published for *Ulva lactuca* cellulose [34,78]. The extracted *Ulva fasciata* cellulose displays the distinguished peaks of the cellulose I allomorph at (2θ) 15.14°, 16.25°, 22.68°, and 34.84° (Figure 14c, blue arrows), which correspond to crystallographic planes of $\bar{1}10$, 110, 200, and 004. Similar observation was reported by Wahlström et al. [34]. The predominance of a peak at (2θ) 22.68 was also reported for Indian *Ulva fasciata* cellulose [9].

The crystallinity index (CrI%) was calculated according to He et al. [79]:

$$CrI\% = \left( \frac{I_{200} - I_{amorphous}}{I_{200}} \right) \times 100 \qquad (9)$$

where $I_{200}$ is the peak intensity of diffraction from 200 lattice plane at (2θ) 22.68°, and $I_{amrophous}$ is the trough intensity attributed to the amorphous portion at (2θ)18° (Figure 14c).

The average crystals size was calculated from Scherrer's equation [79]:

$$\text{Crystal size} = k\,\lambda/\beta\,\cos\theta \qquad (10)$$

where $\beta$ is the full width at half of the maximum of the peak (FWHM), k is the Scherrer's constant, which is equal to 0.91, λ is X-ray wavelength (0.15425 nm), and θ is the Bragg angle in radians.

The interplaner distance between the lattice plans or the d-spacing of the extracted cellulose was also calculated from Bragg's Law [80].

$$d = n\lambda/(2\sin\theta) \qquad (11)$$

where θ is the scattering angle in degrees, n is a positive integer and λ is X-ray wavelength (0.15425 nm).

Thus, the XRD pattern (Figure 14c) proves semi-crystalline polysaccharides with a CrI% of 69.23%, which suggests its recommendable applications in encapsulation and biocomposites. The average crystal size and d-spacing of the extracted cellulose nanofibrils recorded 9.14 nm and 0.47 nm, respectively. The recorded CrI% is better than that reported for *Ulva lactuca* cellulose, which recorded 59% [78] and 48% [34].

However, the crystal size of the extracted Egyptian *Ulva fasciata* cellulose is higher than that reported for the red *Gelidiella aceroso* [81] and brown *Saccharina japonica* algal cellulose [79], which recorded 4.051 nm and 4.57 nm, respectively. However, the d-spacing is lower than that of *Saccharina japonica* algal cellulose, which recorded 0.522 nm [79].

The FESEM micrograph (Figure 15a) of the extracted cellulose reveals a sponge-like surface and a highly cross-linked web-like structure made of several filaments. The EDX analysis (Figure 15b) confirmed the purity of the extracted cellulose, as it is mainly composed of carbon and oxygen with a minor content of nitrogen and some minerals (sodium and magnesium) that recorded approximately 28.74%, 63.87%, 2.22%, 2.3%, and 1.83%, respectively. It also has traces of sulfur, potassium, calcium, and iron, recording approximately 0.54%, 0.14%, 0.13%, and 0.23%, respectively. Thus, the FTIR and FESEM-EDX analyses depicted lignin- and ulvan-free cellulose nanofibrils with approximately 13.88% protein content.

The TGA curve (Figure 16a) shows mass loss of approximately 5.07% from 39 °C to 130 °C, which would be related to the loss of residual moisture. Then, a major loss of approximately 57.78% occurred between 130 °C and 520 °C, followed by an extra, relatively slight weight loss of 19.11% at 900 °C. Similar observation was reported by Lakshmi et al. [9] and Jmel et al. [78] for the thermal decomposition of Indian *Ulva fascita* and *Ulva lactuca* cellulose, respectively, and they attributed that to the carbonization of the polysaccharidic

chains and the cleavage of the C–H and C–C bonds. According to Wang et al. [82], these two stages are related to the degradation of the amorphous and crystalline regions of the extracted cellulose nanofibrils. The DTG curve (Figure 16b) revealed the higher thermal stability of the extracted cellulose relative to the other extracted biopolymers, as it depicted a main thermal decomposition peak at 260 °C, which coincided with that of *Ulva lactuca* cellulose [34,78]. That strengthens its application as a strengthening agent in the preparation of biocomposites and packaging materials. The ash content in the extracted cellulose is approximately 18%, which coincides with the results of the aforementioned EDX analysis depicting a low content of inorganic nutrients. The presence of a tiny peak at 145 °C (Figure 16b) might indicate the presence of minor amounts of hemicellulose [80]. However, the absence of peaks between 380 °C and 500 °C confirms the FTIR results (Figure 14b), denoting the absence of lignin [80]. The DTA (Figure 16c) curve confirms the findings of the TGA (Figure 16a) curve depicting three wide endothermic peaks at 59–120 °C, 130–260 °C, and 690–830 °C. The main endothermic peak $T_p$ 118.54 °C with an enthalpy change ΔH 40.384 J/g appearing in the DSC thermogram (Figure 16d) could be mainly related to the trapped moisture evaporation from the cellulose structure [83].

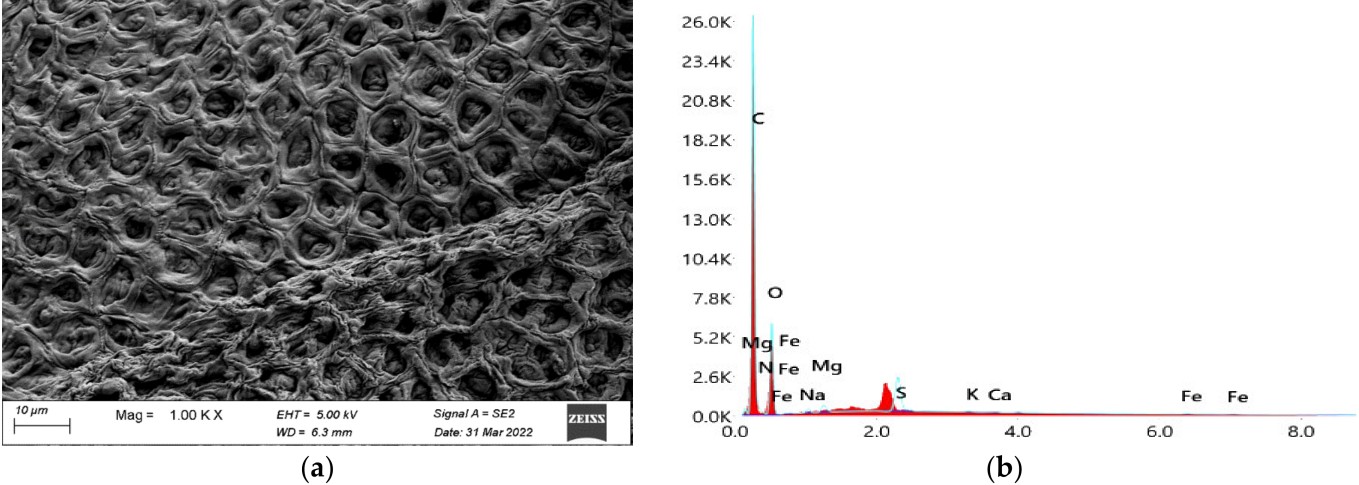

(**a**)

(**b**)

**Figure 15.** FESEM images (**a**) and EDX (**b**) of the extracted *Ulva fasciata* cellulose nanofibrils.

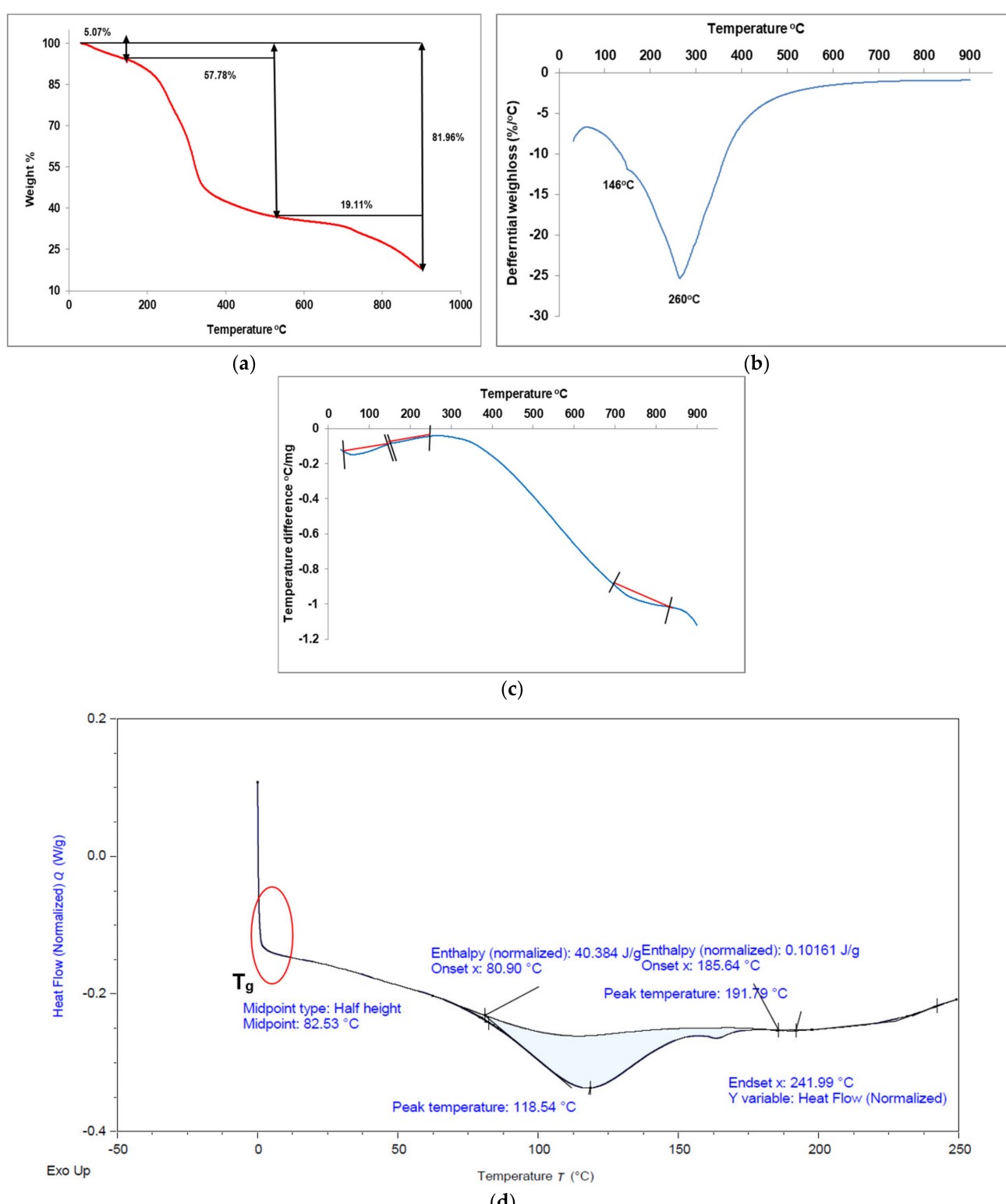

**Figure 16.** The TGA (**a**), DTG (**b**), DTA (**c**), and DSC (**d**) curves of the extracted *Ulva fasciata* cellulose nanofibrils.

## 4. The Obtained Value-Added Products and Their Suggested Possible Applications

The analysis of the collected *Ulva fasciata* biomass revealed total N, $P_2O_5$, and $K_2O$ contents of 3.17% and a C/N ratio of 11.71, which are in accordance with the Egyptian organic fertilizer standard 8079/2017 and promotes its utilization as an organic fertilizer. The capability of *Ulva fasciata* to act as an eco-friendly organic fertilizer was preliminary

tested on two ornamental flowering plants; *Lantana camara* cv. nana yellow (its common name is lantana), and *Hedera helix* (its common name is ivy) and one aromatic medicinal plant; *Mentha piperita* (its common name is peppermint). The fertilizer was prepared at a final concentration of 1000 mg/L, and its biochemical analysis revealed a total dissolved solids concentration of 5000 mg/L, electrical conductivity value of 6.25, a pH of 5.4, organic carbon and organic matter contents of 15.78% and 27.15%, respectively. The algal fertilizer was applied periodically to the plant pots at a rate of 100 mL/week over a total period of two months. The plants in the fertilized pots (Figure 17a) expressed efficient vegetative growth, but the plants in the unfertilized pots lost their leaves and died (Figure 17b). A detailed investigation for its applicability as an organic fertilizer is ongoing now in our lab.

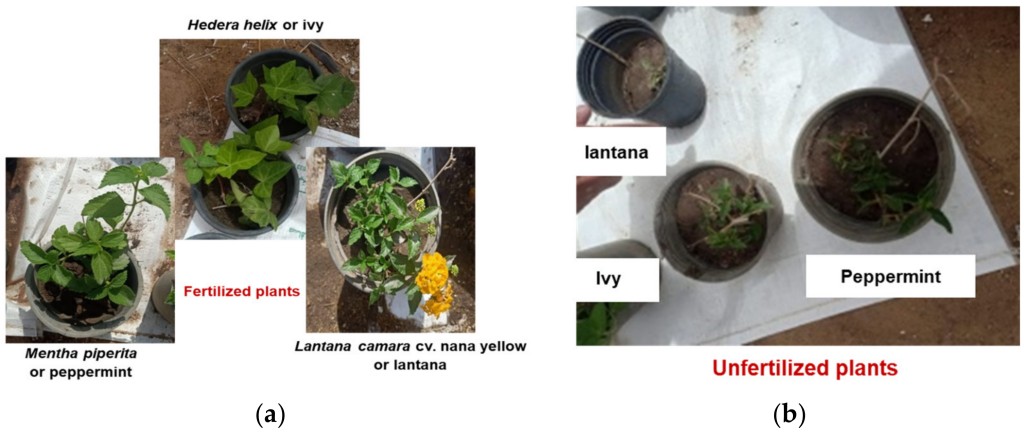

(a)    (b)

**Figure 17.** Preliminary fertilizer experiment using *Ulva faciata*.

The collected *Ulva fasciata* biomass had a calorific value of 15.19 MJ/kg, which is high enough to suggest that it could be used as a solid biofuel. Moreover, the collected *Ulva fasciata* biomass is characterized by high carbohydrate and low lignin contents of approximately 44.85% and 1.5%, respectively, recommending its application in liquid biofuels production (bioethanol and biobutanol) and other biorefineries (acidic and lactic acids).

The sufficient protein, fiber, lipid, and ash contents of the collected *Ulva fasciata*, which recorded approximately 13.13%, 9.9%, 3.27%, and 21%, respectively, and the relatively high concentrations of omega-3 fatty acids (n-3 PUFAs) and omega-9 fatty acids (n-9 MUFAs), besides, the relatively low omega-6 fatty acids (n-6 PUFAs) and a n-6/n-3 ratio of 0.13, also recommend its applicability as food additives and as animal feed.

Additionally, the suggested sequential zero-waste residue process produced different bioproducts that could be used in different industries (Table 10). The eight components that were extracted are: mineral rich water extract (MRWE), chlorophyll$_{a,b}$, carotenoids, starch, lipids, ulvan, proteins, and cellulose, and represent 34.89%, 2.61%, 0.41%, 12.55%, 3.27%, 22.24%, 13.37%, and 10.66% of *Ulva fasciata* dry weight, respectively.

Further, the efficient biocidal activity of the extracted ulvan against pathogenic microorganisms, besides, its pioneering recorded biocidal activity against the corrosive sulfate reducing bacteria, recommends its application for medical purposes, water densification, and mitigation of microbially induced corrosion in the oil and gas industry.

The thorough characterization and identification of pigments, antioxidants, and amino acid profiles, in addition to the cytotoxicity, antioxidant, and anti-cancer effects of the extracts for medical applications, are currently the focus of additional work in our lab. Further research is currently being conducted in our lab to determine the viability of producing bioplastics and membranes from the starch, cellulose, and ulvan produced, as well as probiotics and organic fertilizer from the produced lipids, proteins, and mineral-rich water extract.

**Table 10.** Value-added green and sustainable products obtained from the collected *Ulva fasciata* and some suggested industrial applications.

| Product | Suggested Applications |
|---|---|
| Raw biomass | Animal feeder, biofertilizer, food, and pharmaceutical industries, solid biofuel, bioethanol and biobutanol production |
| Chlorophyll$_{a,b}$ Carotenoids | Coloring agent, food, and pharmaceutical industries |
| Mineral rich water extract | Biofertilizer |
| Starch | Starch-based bioplastics, food, and pharmaceutical industries |
| Lipid | Food supplement, animal feeder, and pharmaceutical industries |
| Ulvan | Biocide, water densification, food packaging, biomedical applications, pharmaceutical industries, and biocorrosion mitigation in oil and gas industry. |
| Protein | Animal feeder, food, and pharmaceutical industries |
| Cellulose | Bioethanol production, food, paper, and pharmaceutical industries |

## 5. Conclusions

The current study outlines an eco-friendly, simple, sequential process for producing a variety of value-added products with different possible industrial applications from the plentiful and easily accessible green *Ulva fasciata*. That suggested sequential zero-waste biomass residue process yielded 34.89% mineral rich water extract (MRWE), 2.61% chlorophyll$_{a,b}$, 0.41% carotenoids, 12.55% starch, 3.27% lipids, 22.24% ulvan, 13.37% proteins, and 10.66% cellulose of *Ulva fasciata* dry weight. Thus, the Egyptian shoreline *Ulva fasciata* is considered an excellent source of lipids, proteins, fibers, and inorganic minerals, as well as carbohydrates and other valuable biopolymers, which are starch, ulvan, and cellulose. This suggested process can be applied in design and construction of viable biorefinery for green products based on marine algal biomass, which would open a new market in Egypt for the bio-based blue economy. It can also be inferred from this study that using *Ulva fasciata* biomass as a sustainable feedstock for producing different biopolymers, organic fertilizer, biocide, and solid biofuel is very strategic and promising given that it only needs three basic inputs: seawater, sunlight, and carbon dioxide. However, it is crucial to stress that the precise costs of *Ulva*-derived products used on a wide scale are not yet known because they have not yet hit the market and their special qualities are unknown. Nevertheless, this suggested zero-waste biomass residue process can, among other things, improve the sustainability of resource use via maximum biomass conversion, mitigate the spread of invasive algal biomass on the Egyptian Mediterranean shoreline, and consequently lessen the negative impact of noxious algal blooms on tourism activities, the ecosystem, and the economy. Yet, much more research is required to assess the commercial viability of the suggested technology, which would depend on the harvesting season, the harvesting cost, the feedstock processing, the marketability of the obtained products, and their applicability to low-carbon industries.

**Author Contributions:** N.Sh.E.-G.: the corresponding author, put forth the idea of scientific research, conceived and designed research, supervised experimental work, validated the results, interpreted and discussed the results, and wrote the manuscript; H.N.N.: collection and sampling of macroalgae, help in designing and conceiving the research, help in validation, curation, interpretation and discussion of obtained data; A.R.I.: conducting, validating, curating, and discussing the biocidal activity; H.R.A. and B.A.A.: performed all the extraction and analysis of the value-added products; K.M.A.: performed the microscopic identification of macroalgae; M.M.: performed all proximate, and biochemical analyses of macroalgae and the preliminary organic fertilizer experiment. All authors have read and agreed to the published version of the manuscript.

**Funding:** This paper is based upon work supported by Science, Technology & Innovation Funding Authority (STDF) under grand number 45614.

**Institutional Review Board Statement:** Not applicable.

**Informed Consent Statement:** Not applicable.

**Data Availability Statement:** Not applicable.

**Acknowledgments:** Authors express their gratitude to the Science, Technology & Innovation Funding Authority (STDF) for funding this work under the grand number 45614.

**Conflicts of Interest:** The authors declare no conflict of interest.

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
