# Peer review of "A Fully Integrated Biorefinery Process for the Valorization of Ulva fasciata into Different Green and Sustainable Value-Added Products"

_sustainability, doi:10.3390/su15097319_

Round 1
Reviewer 1 Report
The authors investigate the process of conversion of Ulva fasciata algae into several value-added products like cellulose, proteins, lipids, ulvan and others via complex extraction by simple solvents (water, ethanol etc.). They thoroughly characterize both initial feedstock and the obtaining products.
On the whole, the article is well written and my only suggestion is to complement Introduction by information concerning previous reports on Ulva fasciata processing into value-added products in more detail, than it is now, in order to reflect the novelty of this work more clearly.
Author Response
- On the whole, the article is well written and my only suggestion is to complement Introduction by information concerning previous reports on Ulva fasciata processing into value-added products in more detail, than it is now, in order to reflect the novelty of this work more clearly.
Thanks a lot for your encouraging comments about the manuscript.
More details about the previous extracted value-added products were added as requested from the respectable reviewer in the revised manuscript.

Reviewer 2 Report
English language usage needs to revise throughout the manuscript
Always give full form during first-time usage in the manuscript (FTIR, TGA, DSC, EDX, SEM, etc.,) or provide nomenclature
Line 145 “All tests were performed in triplicates” Line 176 “All examinations were done in duplicates” why uniformity is not adopted?
Line 196, it must be “tested” not “texted”
Modify Figure 5c to table form
Table 2 ultimate analysis seems to be incomplete without H, N, S, and O content data
It will be better if the analysis/analytical technique used and their findings are reported in text form rather than in sentences eg. TGA-DSC, EDX, FTIR, etc., against their corresponding findings
The macro-algae have a good fatty acid composition in their lipid. Why it has not been suggested for producing bio-diesel or as an additive to diesel fuel? Can look into this prospect
Add quantitative results to the conclusion section to support the claims
Don’t include a table under the conclusion. Move Table 10 to the previous section
Is there any rough estimate of how this concept would impact Egypt’s economics and employment opportunities? Try to include this
Try to link the work with UN sustainable development goals
Overall it is good work.
Author Response
- English language usage needs to revise throughout the manuscript
Revised and modified
- Always give full form during first-time usage in the manuscript (FTIR, TGA, DSC, EDX, SEM, etc.,) or provide nomenclature
All full words are provided
- Line 145 “All tests were performed in triplicates” Line 176 “All examinations were done in duplicates” why uniformity is not adopted?
It was typing mistake and corrected
- Line 196, it must be “tested” not “texted”
Corrected
- Modify Figure 5c to table form
The elemental analysis by EDX is provided in attached table to figure 5c
- Table 2 ultimate analysis seems to be incomplete without H, N, S, and O content data
It is proximate analysis and the table title is corrected
- It will be better if the analysis/analytical technique used and their findings are reported in text form rather than in sentences eg. TGA-DSC, EDX, FTIR, etc., against their corresponding findings
Modified as requested
- The macro-algae have a good fatty acid composition in their lipid. Why it has not been suggested for producing bio-diesel or as an additive to diesel fuel? Can look into this prospect
Further work is ongoing now in our lab to produce biodiesel, biooil, bioethanol, and biochar from Uva fasciata in a full integrated process and will be published soon in another publication.
- Add quantitative results to the conclusion section to support the claims
Done as requested from the respectable reviewer
- Don’t include a table under the conclusion. Move Table 10 to the previous section
Done as requested from the respectable reviewer
- Is there any rough estimate of how this concept would impact Egypt’s economics and employment opportunities? Try to include this
Modified as much as we can
- Try to link the work with UN sustainable development goals
Modified as much as we can
- Overall it is good work.
Thanks a lot for your encouraging comments about the manuscript.

Reviewer 3 Report
In this article, the authors presented a detailed analysis of the biorefinery process for the valorization of Ulva fasciata into value-added products. By different kinds of extraction solvents and processes, different value-added products like pigments, mineral-rich water, starch, lipid, ulvan, protein, and cellulose could be extracted from the biomass efficiently. And all the extraction fractions were analyzed systematically and compared with peers. But the highlight of this research project could be improved to produce more scientific insight. Low selectivity and lack of breakthrough extraction or conversion technology are the key components that needs to be addressed. I recommend this article be reconsidered after major revision.
1. In the sentence “Valorization of such wasted green macroalgae into different value-added products and sustainable biopolymers has recently attracted a lot of interest”, please indicate some of the example utilizations. Line 53
2. In the last paragraph of introduction, please provide more information regarding experiment design and research interests.
3. The ultimate analysis of the algal biomass has been performed at General Organization for Export and Import Control (GOEIC), that included measuring the moisture content (EN15414-3/2011), volatile matter (EN15402/2011), ash content (EN15403/2011), and calorific value (ASTM D5865-19). The ultimate is analysis is analysis that, also called element analysis, which could be performed by CHNS. Moisture, volatile, fixed carbon, and ash content is called proximate analysis. Please double-check and perform a suitable analysis.
4. For pigments and antioxidants quantification, a GC-MS-FID system could provide more accurate results.
5. All the test results are comparing to other research articles and get a comparable result. Please indicate the highlight or breakthrough of this research project.
6. The overall process did not show significant state-of-the-art conversion technology to produce value-added chemical. More insights are needed other than solvent extractions. The selectivity of value-added chemicals is also limited. Economic value needs to be evaluated to prove this process could be “value-added” or not.
Author Response
In this article, the authors presented a detailed analysis of the biorefinery process for the valorization of Ulva fasciata into value-added products. By different kinds of extraction solvents and processes, different value-added products like pigments, mineral-rich water, starch, lipid, ulvan, protein, and cellulose could be extracted from the biomass efficiently. And all the extraction fractions were analyzed systematically and compared with peers. But the highlight of this research project could be improved to produce more scientific insight. Low selectivity and lack of breakthrough extraction or conversion technology are the key components that needs to be addressed. I recommend this article be reconsidered after major revision.
- In the sentence “Valorization of such wasted green macroalgae into different value-added products and sustainable biopolymers has recently attracted a lot of interest”, please indicate some of the example utilizations. Line 53
Added as requested for the respectable reviewer
- In the last paragraph of introduction, please provide more information regarding experiment design and research interests.
Modified as requested for the respectable reviewer
- The ultimate analysis of the algal biomass has been performed at General Organization for Export and Import Control (GOEIC), that included measuring the moisture content (EN15414-3/2011), volatile matter (EN15402/2011), ash content (EN15403/2011), and calorific value (ASTM D5865-19). The ultimate is analysis is analysis that, also called element analysis, which could be performed by CHNS. Moisture, volatile, fixed carbon, and ash content is called proximate analysis. Please double-check and perform a suitable analysis.
It is proximate analysis and the table title is corrected
- For pigments and antioxidants quantification, a GC-MS-FID system could provide more accurate results.
Further work is undertaken now in our lab for full characterization and identification of pigments, antioxidants, amino acids profile, besides the cytotoxicity, antioxidant, and anti-cancer activities of the extracts and the results will be published in another publication, as soon as we finalize it.
- All the test results are comparing to other research articles and get a comparable result. Please indicate the highlight or breakthrough of this research project.
Explained as much as we can in the revised manuscript
- The overall process did not show significant state-of-the-art conversion technology to produce value-added chemical. More insights are needed other than solvent extractions. The selectivity of value-added chemicals is also limited. Economic value needs to be evaluated to prove this process could be “value-added” or not.
Further work is ongoing now in our lab to study the feasibility of producing bioplastics and membranes using the produced starch, ulvan, and cellulose, besides bioethanol from the produced starch and cellulose, in addition to probiotics and organic fertilizer using the produced lipids, proteins, and mineral-rich water extract.

Reviewer 4 Report
The submitted manuscript (ID: sustainability-2321800) reported an eco-friendly fully integrated biorefinery process for marine Ulva fasciata macroalgae. The authors used some analytical tools to show this work would serve as a solution for ecosystem bioremediation, effective utilization of marine macroalgal resources, and a new initiative to promote a green and low-carbon economy. However, this manuscript still has some problems and needs revisions to meet the requirements of the journal. Some questions are as follows:
1. In the introduction section, some more descriptions of biomass research status should be included, and the latest literature is suggested, e.g., Bioresource Technology 2023, 369, 128390; ChemSusChem 2022, 15, e202102581; Green Chemistry, 2021, 23, 6675-6697; Nature Communications, 2019, 10, 699.
2. There are some images in the article that depict content that is not very relevant to the conclusions of the article, for example, microscopic examination and surface view through the middle region of the collected Ulva fasciata thallus in Figure 4 depict content that is not very meaningful and the reader may prefer to see a surface view of the raw material after treatment.
3. This article is tiring to read, probably because the experimental part does not flow smoothly with the conclusion. The various steps of the experiment are summarised in the previous article, and the significance of each of them is not well explained later in the article, with the possibility of simply piling up the workload.
4. The purity analysis of cellulose, trace elements, and proteins was carried out in the experimental section and was not compared with other work, thus not highlighting the advantages of this work. There are many more problems such as these which are not listed here.
5. Most of the figures in the article are not well labeled for the reader, including but not limited to Figure 7(c), which simply identifies the location and does not explain it well.
6. The article is dense with grammatical errors and confusing formatting, and it is hoped that the author will take note of the formatting and grammatical issues and correct them.
Author Response
The submitted manuscript (ID: sustainability-2321800) reported an eco-friendly fully integrated biorefinery process for marine Ulva fasciata macroalgae. The authors used some analytical tools to show this work would serve as a solution for ecosystem bioremediation, effective utilization of marine macroalgal resources, and a new initiative to promote a green and low-carbon economy. However, this manuscript still has some problems and needs revisions to meet the requirements of the journal. Some questions are as follows:
- In the introduction section, some more descriptions of biomass research status should be included, and the latest literature is suggested, e.g., Bioresource Technology 2023, 369, 128390; ChemSusChem 2022, 15, e202102581; Green Chemistry, 2021, 23, 6675-6697; Nature Communications, 2019, 10, 699.
Huang J, Wang J, Huang Z, Liu T, Li H. Photothermal technique-enabled ambient production of microalgae biodiesel: Mechanism and life cycle assessment. Bioresour Technol 2023;369:128390. https://doi.org/10.1016/j.biortech.2022.128390
Ye Meng, Song Yang, Hu Li. Electro- and photocatalytic oxidative upgrading of bio-based 5-hydroxymethylfurfural. ChemSusChem 2022. https://doi.org/10.1002/cssc.202102581
Hongguo Wu, Hu Li, Zhen Fang. Hydrothermal amination of biomass to nitrogenous chemicals. Green Chem., 2021, 23: 6675. https://doi.org/10.1039/d1gc02505h
Li, H., Guo, H., Su, Y. et al. N-formyl-stabilizing quasi-catalytic species afford rapid and selective solvent-free amination of biomass-derived feedstocks. Nat Commun 10, 699 (2019). https://doi.org/10.1038/s41467-019-08577-4
Kindly be noted that the listed suggested references are not related to the manuscript.
So plz accept our apology for not adding them to the manuscript.
- There are some images in the article that depict content that is not very relevant to the conclusions of the article, for example, microscopic examination and surface view through the middle region of the collected Ulva fasciata thallus in Figure 4 depict content that is not very meaningful and the reader may prefer to see a surface view of the raw material after treatment.
We have reached to zero-waste and there is no left residue to photograph or examine under microscope.
The performed examination was done to ascertain the primary visual and microscopic characteristics of Ulva fasciata.
- This article is tiring to read, probably because the experimental part does not flow smoothly with the conclusion. The various steps of the experiment are summarised in the previous article, and the significance of each of them is not well explained later in the article, with the possibility of simply piling up the workload.
Modified as much as we can
- The purity analysis of cellulose, trace elements, and proteins was carried out in the experimental section and was not compared with other work, thus not highlighting the advantages of this work. There are many more problems such as these which are not listed here.
That is covered in the revised manuscript
- Most of the figures in the article are not well labeled for the reader, including but not limited to Figure 7(c), which simply identifies the location and does not explain it well.
That is modified in the revised manuscript
- The article is dense with grammatical errors and confusing formatting, and it is hoped that the author will take note of the formatting and grammatical issues and correct them.
Revised and modified

Round 2
Reviewer 3 Report
The author has responded accordingly to all my comments and suggestions. I recommend this article be accepted in its current format.
Reviewer 4 Report
The revised manuscript has been improved and can be accepted for publication.